# An equivalence between high dimensional Bayes optimal inference and M-estimation

**Madhu Advani**      **Surya Ganguli**
Department of Applied Physics, Stanford University
msadvani@stanford.edu   and   sganguli@stanford.edu

## Abstract

When recovering an unknown signal from noisy measurements, the computational difficulty of performing optimal Bayesian MMSE (minimum mean squared error) inference often necessitates the use of maximum a posteriori (MAP) inference, a special case of regularized M-estimation, as a surrogate. However, MAP is suboptimal in high dimensions, when the number of unknown signal components is similar to the number of measurements. In this work we demonstrate, when the signal distribution and the likelihood function associated with the noise are both log-concave, that optimal MMSE performance is asymptotically achievable via another M-estimation procedure. This procedure involves minimizing convex loss and regularizer functions that are *nonlinearly smoothed* versions of the widely applied MAP optimization problem. Our findings provide a new heuristic derivation and interpretation for recent optimal M-estimators found in the setting of linear measurements and additive noise, and further extend these results to nonlinear measurements with non-additive noise. We numerically demonstrate superior performance of our optimal M-estimators relative to MAP. Overall, at the heart of our work is the revelation of a remarkable equivalence between two seemingly very different computational problems: namely that of high dimensional Bayesian integration underlying MMSE inference, and high dimensional convex optimization underlying M-estimation. In essence we show that the former difficult integral may be computed by solving the latter, simpler optimization problem.

## 1   Introduction

Modern technological advances now enable scientists to simultaneously record hundreds or thousands of variables in fields ranging from neuroscience and genomics to health care and economics. For example, in neuroscience, we can simultaneously record $P = O(1000)$ neurons in behaving animals. However, the number of measurements $N$ we can make of these $P$ dimensional neural activity patterns can be limited in any given experimental condition due to constraints on recording time. Thus a critical parameter is the measurement density $\alpha = \frac{N}{P}$. Classical statistics focuses on the limit of few variables and many measurements, so $P$ is finite, $N$ is large, and $\alpha \to \infty$. Here, we instead consider the modern high dimensional limit where the measurement density $\alpha$ remains finite as $N, P \to \infty$. In this important limit, we ask what is the optimal way to recover signal from noise?

More precisely, we wish to recover an unknown signal vector $\mathbf{s}^0 \in \mathbb{R}^P$ given $N$ noisy measurements

$$y_\mu = r(\mathbf{x}_\mu \cdot \mathbf{s}^0, \epsilon_\mu) \quad \text{where} \quad \mathbf{x}_\mu \in \mathbb{R}^P \quad \text{and} \quad y_\mu \in \mathbb{R}, \quad \text{for} \quad \mu = 1, \dots, N. \tag{1}$$

Here, $\mathbf{x}_\mu$ and $y_\mu$ are input-output pairs for measurement $\mu$, $r$ is a measurement nonlinearity, and $\epsilon_\mu$ is a noise realization. For example, in a brain machine interface, $\mathbf{x}_\mu$ could be a neural activity pattern, $y_\mu$ a behavioral covariate, and $\mathbf{s}^0$ the unknown regression coefficients of a decoder relating neural activity to behavior. Alternatively, in sensory neuroscience, $\mathbf{x}_\mu$ could be an external stimulus,

$y_\mu$ a single neuron's response to that stimulus, and $\mathbf{s}^0$ the unknown receptive field relating stimulus to neural response. We assume the noise $\epsilon_\mu$ is independent and identically distributed (iid) across measurements, implying the outputs $y_\mu$ are drawn iid from a noise distribution $P_{y|z}(y_\mu|z_\mu)$, where $z_\mu = \mathbf{x}_\mu \cdot \mathbf{s}^0$. Similarly, we assume the signal components $s_i^0$ are drawn iid from a prior signal distribution $P_s(s^0)$. We denote its variance below by $\sigma_s^2$. Finally, we denote by $\mathbf{X} \in \mathbb{R}^{N \times P}$ the input or measurement matrix, whose $\mu$'th row is $\mathbf{x}_\mu$, and by $\mathbf{y} \in \mathbb{R}^N$ the measurement output vector whose $\mu$'th component is $y_\mu$. In this paper, we will focus on the case of dense iid random Gaussian measurements, normalized so that $\langle \mathbf{x}_\mu \cdot \mathbf{x}_\nu \rangle = \gamma \delta_{\mu,\nu}$. In the case of systems identification in sensory neuroscience, this choice would correspond to an oft used white noise stimulus at contrast $\gamma$.

Now given measurement data $(\mathbf{X}, \mathbf{y})$, as well as knowledge of the nonlinearity $r(\cdot)$ and the signal $P_s$ and noise $P_{y|z}$ distributions, what is the best way to infer an estimate $\hat{\mathbf{s}}$ of the unknown signal $\mathbf{s}^0$? We characterize the performance of an estimate $\hat{\mathbf{s}}$ by its mean squared error (MSE), $\|\hat{\mathbf{s}} - \mathbf{s}^0\|_2^2$, averaged over noise realizations and measurements. The best minimal MSE (MMSE) estimator is given by optimal Bayesian integration to compute the posterior mean:

$$\hat{\mathbf{s}}^{\text{MMSE}} = \int \mathbf{s}\, P(\mathbf{s}|\mathbf{X}, \mathbf{y})\, d\mathbf{s}. \tag{2}$$

Unfortunately, this integral is generally intractable in high dimensions, at large $P$; both numerical integration and Monte Carlo methods for estimating the integral require computational time growing exponentially in $P$ for high accuracy. Consequently, an often used surrogate for MMSE inference is maximum a posteriori (MAP) inference, which computes the mode rather than the mean of the posterior distribution. Thus MAP relies on optimization rather than integration:

$$\hat{\mathbf{s}}^{\text{MAP}} = \arg\max_{\mathbf{s}} P(\mathbf{s}|\mathbf{X}, \mathbf{y}) = \arg\min_{\mathbf{s}} [-\log P(\mathbf{s}|\mathbf{X}, \mathbf{y})]. \tag{3}$$

Assuming inputs $\mathbf{X}$ are independent of the unkown signal $\mathbf{s}^0$, the above expression becomes

$$\hat{\mathbf{s}}^{\text{MAP}} = \arg\min_{\mathbf{s}} \left[ \sum_{\mu=1}^{N} -\log P_{y|z}(y_\mu|\mathbf{x}_\mu \cdot \mathbf{s}) + \sum_{i=1}^{P} -\log P_s(s_i) \right]. \tag{4}$$

A related algorithm is maximum likelihood (ML), which seeks to maximize the likelihood of the data given a candidate signal $\mathbf{s}$. ML is equivalent to MAP in (4) but without the second sum, i.e. without prior information on the signal.

While ML is typically optimal amongst unbiased estimators in the classical statistical limit $\alpha \to \infty$ (see e.g. [1]), neither MAP nor ML are optimal in high dimensions, at finite $\alpha$. Therefore, we consider a broader class of estimators known as regularized M-estimators, corresponding to the optimization problem

$$\hat{\mathbf{s}} = \arg\min_{s} \left[ \sum_{\mu=1}^{N} \mathcal{L}(y_\mu, \mathbf{x}_\mu \cdot \mathbf{s}) + \sum_{i=1}^{P} \sigma(s_i) \right]. \tag{5}$$

Here $\mathcal{L}(y, \eta)$ is a *loss function* and $\sigma$ is a *regularizer*. We assume both to be convex functions in $\eta$ and $s$ respectively. Note that MAP inference corresponds to the choice $\mathcal{L}(y, \eta) = -\log P_{y|z}(y|\eta)$ and $\sigma(s) = -\log P_s(s)$. ML inference corresponds to the same loss function but without regularization: $\sigma(s) = 0$. Other well known M-estimators include LASSO [2], corresponding to the choice $\mathcal{L}(y, \eta) = \frac{1}{2}(y - \eta)^2$ and $\sigma(s) \propto |s|$, or the elastic net [3], which includes an addition quadratic term on the LASSO regularizer. Such M-estimators are heuristically motivated as a convex relaxation of MAP inference for sparse signal distributions, and have been found to be very useful in such settings. However, a general theory for how to select the optimal M-estimator in (5) given the generative model of data in (1) remains elusive. This is the central problem we address in this work.

## 1.1 Related work and Outline

Seminal work [4] found the optimal *unregularized* M-estimator using variational methods in the special case of linear measurements and additive noise, i.e. $r(z, \epsilon) = z + \epsilon$ in (1). In this same setting, [5] characterized unregularized M-estimator performance via approximate message passing (AMP) [6]. Following this, the performance of regularized M-estimators in the linear additive setting was characterized in [7], using non-rigorous statistical physics methods based on replica theory, and

in [8], using rigorous methods different from [4, 5]. Moreover, [7] found the optimal regularized M-estimator and demonstrated, surprisingly, zero performance gap relative to MMSE. The goals of this paper are to (1) interpret and extend previous work by deriving an equivalence between optimal M-estimation and Bayesian MMSE inference via AMP and (2) to derive the optimal M-estimator in the more general setting of nonlinear measurements and non-additive noise.

To address these goals, we begin in section 2 by describing a pair of AMP algorithms, derived heuristically via approximations of belief propagation (BP). The first algorithm, mAMP, is designed to solve M-estimation in (5), while the second, bAMP, is designed to solve Bayesian MMSE inference in (2). In section 3 we derive a connection, via AMP, between M-estimation and MMSE inference: we find, for a particular choice of optimal M-estimator, that mAMP and bAMP have the same fixed points. To quantitatively determine the optimal M-estimator, which depends on some smoothing parameters, we must quantitatively characterize the performance of AMP, which we do in section 4. We thereby recover optimal M-estimators found in recent works in the linear additive setting, without using variational methods, and moreover find optimal M-estimators in the nonlinear, non-additive setting. Our non-variational approach through AMP also provides an intuitive explanation for the form of the optimal M-estimator in terms of Bayesian inference. Intriguingly, the optimal M-estimator resembles a smoothed version of MAP, with lower measurement density requiring more smoothing. In Section 4, we also demonstrate, through numerical simulations, a substantial performance improvement in inference accuracy achieved by the optimal M-estimator over MAP under nonlinear measurements with non-additive noise. We end with a discussion in section 5.

## 2 Formulations of Bayesian inference and M-estimation through AMP

Both mAMP and bAMP, heuristically derived in the supplementary material [1] (SM) sections 2.2-2.4 though approximate BP applied to (5) and (2) respectively, can be expressed as special cases of a generalized AMP (gAMP) algorithm [9], which we first describe. gAMP is a set of iterative equations,

$$\boldsymbol{\eta}^t = \mathbf{X}\hat{\mathbf{s}}^t + \lambda_\eta^t G_y(\lambda_\eta^{t-1}, \mathbf{y}, \boldsymbol{\eta}^{t-1}) \qquad\qquad \hat{\mathbf{s}}^{t+1} = G_s\left(\lambda_h^t, \hat{\mathbf{s}}^t - \lambda_h^t \mathbf{X}^T G_y(\lambda_\eta^t, \mathbf{y}, \boldsymbol{\eta}^t)\right) \qquad (6)$$

$$\lambda_h^t = \left(\frac{\gamma\alpha}{N} \sum_{\nu=1}^N \frac{\partial}{\partial\eta} G_y(\lambda_\eta^t, y_\nu, \eta_\nu^t)\right)^{-1} \qquad \lambda_\eta^{t+1} = \frac{\gamma\lambda_h^t}{P} \sum_{j=1}^P \frac{\partial}{\partial h} G_s(\lambda_h^t, \hat{s}_j^t - \lambda_h^t \mathbf{X}_j^T G_y(\lambda_\eta^t, \mathbf{y}, \boldsymbol{\eta}^t)), \tag{7}$$

that depend on the scalar functions $G_y(\lambda_\eta, y, \eta)$ and $G_s(\lambda_h, h)$ which, in our notation, act component-wise on vectors so that $\mu^{\text{th}}$ component $G_y(\lambda_\eta, \mathbf{y}, \boldsymbol{\eta})_\mu = G_y(\lambda_\eta, y_\mu, \eta_\mu)$ and the $i^{\text{th}}$ component $G_s(\lambda_h, \mathbf{h})_i = G_s(\lambda_h, h_i)$. Initial conditions are given by $\hat{\mathbf{s}}^{t=0} \in \mathbb{R}^P$, $\lambda_\eta^{t=0} \in \mathbb{R}^+$ and $\boldsymbol{\eta}^{t=-1} \in \mathbb{R}^N$.

Intuitively, one can think of $\boldsymbol{\eta}^t$ as related to the linear part of the measurement outcome predicted by the current guess $\hat{\mathbf{s}}^t$, and $G_y$ is a measurement correction map that uses the actual measurement data $\mathbf{y}$ to correct $\boldsymbol{\eta}^t$. Also, intuitively, we can think of $G_s$ as taking an input $\hat{\mathbf{s}}^t - \lambda_h^t \mathbf{X}^T G_y(\lambda_\eta^t, \mathbf{y}, \boldsymbol{\eta}^t)$, which is a measurement based correction to $\hat{\mathbf{s}}^t$, and yielding as output a further, measurement independent correction $\hat{\mathbf{s}}^{t+1}$, that could depend on either a regularizer or prior. We thus refer to the functions $G_y$ and $G_s$ as the measurement and signal correctors respectively. gAMP is thus alternating measurement and signal correction, with time dependent parameters $\lambda_h^t$ and $\lambda_\eta^t$. These equations were described in [9], and special cases of them were studied in various works (see e.g. [5, 10]).

### 2.1 From M-estimation to mAMP

Now, applying approximate BP to (5) when the input vectors $\mathbf{x}_\mu$ are iid Gaussian, again with normalization $\langle \mathbf{x}_\mu \cdot \mathbf{x}_\mu \rangle = \gamma$, we find (SM Sec. 2.3) that the resulting mAMP equations are a special case of the gAMP equations, where the functions $G_y$ and $G_s$ are related to the loss $\mathcal{L}$ and regularizer $\sigma$ through

$$G_y^M(\lambda_\eta, y, \eta) = \mathcal{M}_{\lambda_\eta}[\mathcal{L}(y, \cdot)]'(\eta), \qquad G_s^M(\lambda_h, h) = \mathcal{P}_{\lambda_h}[\sigma](h). \tag{8}$$

The functional mappings $\mathcal{M}$ and $\mathcal{P}$, the Moreau envelope and proximal map [11], are defined as

$$\mathcal{M}_\lambda[\,f\,](x) = \min_y \left[\frac{(x-y)^2}{2\lambda} + f(y)\right], \qquad \mathcal{P}_\lambda[\,f\,](x) = \arg\min_y \left[\frac{(x-y)^2}{2\lambda} + f(y)\right]. \qquad (9)$$

The proximal map maps a point $x$ to another point that minimizes $f$ while remaining close to $x$ as determined by a scale $\lambda$. This can be thought of as a *proximal descent step* on $f$ starting from $x$ with step length $\lambda$. Perhaps the most ubiquitous example of a proximal map occurs for $f(z) = |z|$, in which case the proximal map is known as the soft thresholding operator and takes the form $\mathcal{P}_\lambda[\,f\,](x) = 0$ for $|x| \le \lambda$ and $\mathcal{P}_\lambda[\,f\,](x) = x - \text{sign}(x)\lambda$ for $|x| \ge \lambda$. This soft thresholding is prominent in AMP approaches to compressed sensing (e.g. [10]). The Moreau envelope is a minimum convolution of $f$ with a quadratic, and as such, $\mathcal{M}_\lambda[\,f\,](x)$ is a smoothed lower bound on $f$ with the same minima [11]. Moreover, differentiating $\mathcal{M}$ with respect to $x$ yields [11] the relation

$$\mathcal{P}_\lambda[\,f\,](x) = x - \lambda \mathcal{M}_\lambda[\,f\,]'(x). \qquad (10)$$

Thus a proximal descent step on $f$ is equivalent to a gradient descent step on the Moreau envelope of $f$, with the same step length $\lambda$. This equality is also useful in proving (SM Sec. 2.1) that the fixed points of mAMP satisfy

$$\mathbf{X}^T \frac{\partial}{\partial \eta} \mathcal{L}(\mathbf{y}, \mathbf{X}\hat{\mathbf{s}}) + \sigma'(\hat{\mathbf{s}}) = \mathbf{0}. \qquad (11)$$

Thus fixed points of mAMP are local minima of M-estimation in (5).

To develop intuition for the mAMP algorithm, we note that the $\hat{\mathbf{s}}$ update step in (6) is similar to the more intuitive proximal gradient descent algorithm [11] which seeks to solve the M-estimation problem in (5) by alternately performing a gradient descent step on the loss term and a proximal descent step on the regularization term, both with the same step length. Thus one iteration of gradient descent on $\mathcal{L}$ followed by proximal descent on $\sigma$ in (5), with both steps using step length $\lambda_h$, yields

$$\hat{\mathbf{s}}^{t+1} = \mathcal{P}_{\lambda_h}[\,\sigma\,](\hat{\mathbf{s}}^t - \lambda_h \mathbf{X}^T \frac{\partial}{\partial \eta} \mathcal{L}(\mathbf{y}, \mathbf{X}\hat{\mathbf{s}}^t)). \qquad (12)$$

By inserting (8) into (6)-(7), we see that mAMP closely resembles proximal gradient descent, but with three main differences: 1) the loss function is replaced with its Moreau envelope, 2) the loss is evaluated at $\boldsymbol{\eta}^t$ which includes an additional memory term, and 3) the step size $\lambda_h^t$ is time dependent. Interestingly, this additional memory term and step size evolution has been found to speed up convergence relative to proximal gradient descent in certain special cases, like LASSO [10].

In summary, in mAMP the measurement corrector $G_y$ implements a gradient descent on the Moreau smoothed loss, while the signal corrector $G_s$ implements a proximal descent step on the regularizer. But because of (10), this latter step can also be thought of as a gradient descent step on the Moreau smoothed regularizer. Thus overall, the mAMP approach to M-estimation is intimately related to Moreau smoothing of both the loss and regularizer.

## 2.2 From Bayesian integration to bAMP

Now, applying approximate BP to (2) when again the input vectors $\mathbf{x}_\mu$ are iid Gaussian, we find (SM Sec. 2.2) that the resulting bAMP equations are a special case of the gAMP equations, where the functions $G_y$ and $G_s$ are related to the noise $P_{y|z}$ and signal $P_s$ distributions through

$$G_y^B(\lambda_\eta, y, \eta) = -\frac{\partial}{\partial \eta} \log\left(P_y(y|\eta, \lambda_\eta)\right), \qquad G_s^B(\lambda_h, h) = \hat{s}^{\text{mmse}}(\lambda_h, h), \qquad (13)$$

where

$$P_y(y|\eta, \lambda) \propto \int P_{y|z}(y|z) e^{-\frac{(\eta - z)^2}{2\lambda}} dz, \qquad \hat{s}^{\text{mmse}}(\lambda, h) = \frac{\int s P_s(s) e^{-\frac{(s-h)^2}{2\lambda}} ds}{\int P_s(s) e^{-\frac{(s-h)^2}{2\lambda}} ds}, \qquad (14)$$

as derived in SM section 2.2. Here $P_y(y|\eta, \lambda)$ is a convolution of the likelihood with a Gaussian of variance $\lambda$ (normalized so that it is a probability density in $y$) and $\hat{s}^{\text{mmse}}$ denotes the posterior mean $\langle s^0|h \rangle$ where $h = s^0 + \sqrt{\lambda}w$ is a corrupted signal, $w$ is a standard Gaussian random variable, and $s^0$ is a random variable drawn from $P_s$.

Inserting these equations into (6)-(7), we see that bAMP performs a measurement correction step through $G_y$ that corresponds to a gradient descent step on the negative log of a Gaussian-smoothed likelihood function. The subsequent signal correction step through $G_s$ is simply the computation of a posterior mean, assuming the input is drawn from the prior and corrupted by additive Gaussian noise with a time-dependent variance $\lambda_h^t$.

## 3    An AMP equivalence between Bayesian inference and M-estimation

In the previous section, we saw intriguing parallels between mAMP and bAMP, both special cases of gAMP. While mAMP performs its measurement and signal correction through a gradient descent step on a Moreau smoothed loss and a Moreau smoothed regularizer respectively, bAMP performs its measurement correction through a gradient descent step on the minus log of a Gaussian smoothed likelihood, and its signal correction though an MMSE estimation problem. These parallels suggest we may be able to find a loss $\mathcal{L}$ and regularizer $\sigma$ such that the corresponding mAMP becomes equivalent to bAMP. If so, then assuming the correctness of bAMP as a solution to (2), the resulting $\mathcal{L}^{opt}$ and $\sigma^{opt}$ will yield the optimal mAMP dynamics, achieving MMSE inference.

By comparing (8) and (13), we see that bAMP and mAMP will have the same $G_y$ if the Moreau-smoothed loss equals the minus log of the Gaussian-smoothed likelihood function:

$$\mathcal{M}_{\lambda_\eta}[\,\mathcal{L}^{\text{opt}}(y, \cdot)\,](\eta) = -\log\left(P_y(y|\eta, \lambda_\eta)\right). \tag{15}$$

Before describing how to invert the above expression to determine $\mathcal{L}^{\text{opt}}$, we would also like to find a relation between the two signal correction functions $G_s^M$ and $G_s^B$. This is a little more challenging because the former implements a proximal descent step while the latter implements an MMSE posterior mean computation. However, we can express the MMSE computation as gradient ascent on the log of a Gaussian smoothed signal distribution (see SM):

$$\hat{s}^{\text{mmse}}(\lambda_h, h) = h + \lambda_h \frac{\partial}{\partial h}\log\left(P_s(h, \lambda_h)\right), \qquad P_s(h, \lambda) \propto \int P_s(s) e^{-\frac{(s-h)^2}{2\lambda}}ds. \tag{16}$$

Moreover, by applying (10) to the definition of $G_s^M$ in (8), we can write $G_s^M$ as gradient descent on a Moreau smoothed regularizer. Then, comparing these modified forms of $G_s^B$ with $G_s^M$, we find a similar condition for $\sigma^{\text{opt}}$, namely that its Moreau smoothing should equal the minus log of the Gaussian smoothed signal distribution:

$$\mathcal{M}_{\lambda_h}[\,\sigma^{\text{opt}}\,](h) = -\log\left(P_s(h, \lambda_h)\right). \tag{17}$$

Our goal is now to compute the optimal loss and regularizer by inverting the Moreau envelope relations (15, 17) to solve for $\mathcal{L}^{\text{opt}}, \sigma^{\text{opt}}$. A sufficient condition [4] to invert these Moreau envelopes to determine the optimal mAMP dynamics is that $P_y(y|z)$ and $P_s(s)$ are log concave with respect to $z$ and $s$ respectively. Under this condition the Moreau envelope will be invertible via the relation $\mathcal{M}_q[-\mathcal{M}_q[-f](\cdot)](\cdot) = f(\cdot)$ (see SM Appendix A.3 for a derivation), which yields:

$$\mathcal{L}^{\text{opt}}(y, \eta) = -\mathcal{M}_{\lambda_\eta}[\log\left(P_y(y|\cdot, \lambda_\eta)\right)](\eta), \qquad \sigma^{\text{opt}}(h) = -\mathcal{M}_{\lambda_h}[\log\left(P_s(\cdot, \lambda_h)\right)](h). \tag{18}$$

This optimal loss and regularizer form resembles smoothed MAP inference, with $\lambda_\eta$ and $\lambda_h$ being scalar parameters that modify MAP through both Gaussian and Moreau smoothing. An example of such a family of smoothed loss and regularizer functions is given in Fig. 1 for the case of a logistic output channel with Laplacian distributed signal. Additionally, one can show that the optimal loss and regularizer are convex when the signal and noise distributions are log-concave. Overall, this analysis yields a dynamical equivalence between mAMP and bAMP as long as at each iteration time $t$, the optimal loss and regularizer for mAMP are chosen through the smoothing operation in (18), but using time-dependent smoothing parameters $\lambda_\eta^t$ and $\lambda_h^t$ whose evolution is governed by (7).

## 4    Determining optimal smoothing parameters via state evolution of AMP

In the previous section, we have shown that mAMP and bAMP have the same dynamics, as long as, at each iteration $t$ of mAMP, we choose a *time dependent* optimal loss $\mathcal{L}_t^{\text{opt}}$ and regularizer $\sigma_t^{\text{opt}}$ through (18), where the time dependence is inherited from the time dependent smoothing parameters $\lambda_\eta^t$ and $\lambda_h^t$. However, mAMP was motivated as an algorithmic solution to the M-estimation problem in (5) for a *fixed* loss and regularizer, while bAMP was motivated as a method of performing the Bayesian integral in (2). This then raises the question, is there a fixed, optimal choice of $\mathcal{L}^{\text{opt}}$ and $\sigma^{\text{opt}}$ in (5) such the corresponding M-estimation problem yields the same answer as the Bayesian integral in (2)?

The answer is yes: simply choose a fixed $\mathcal{L}^{\text{opt}}$ and $\sigma^{\text{opt}}$ through (18) where the smoothing parameters $\lambda_\eta$ and $\lambda_h$ are chosen to be those found at the fixed points of bAMP. To see this, note that fixed points of mAMP with time dependent choices of $\mathcal{L}_t^{\text{opt}}$ and $\sigma_t^{\text{opt}}$ are equivalent to the minima of the

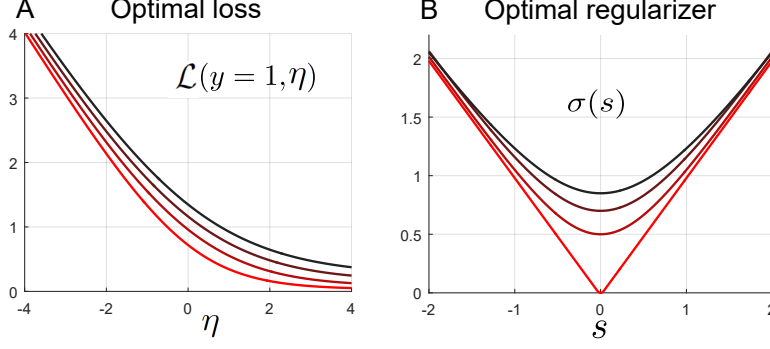

Figure 1: Here we plot the optimal loss (A) and regularizer (B) in (18), for a logistic output $y \in \{0,1\}$ with $P_{y|z}(y=1|z) = \frac{1}{1+e^{-z}}$, and Laplacian signal $s$ with $P_s(s) = \frac{1}{2}e^{-|s|}$. In (A) we plot the loss for the measurement $y = 1$: $\mathcal{L}^{\mathrm{opt}}(y=1, \cdot)$. Both sets of curves from red to black (and bottom to top) correspond to smoothing parameters $\lambda_\eta = (0, 2, 4, 6)$ in (A) and $\lambda_h = (0, 1/2, 1, 2)$ in (B). With zero smoothing, the red curves at the bottom correspond to the MAP loss and regularizer.

M-estimation problem in (5), with the choice of loss and regularizer that this time dependent sequence converges to: $\mathcal{L}^{\mathrm{opt}}_\infty$ and $\sigma^{\mathrm{opt}}_\infty$ (this follows from an extension of the argument that lead to (11)). In turn the fixed points of mAMP are equivalent to those of bAMP under the choice (18). These equivalences then imply that, if the bAMP dynamics for $(\hat{\mathbf{s}}^{\mathbf{t}}, \lambda_\eta^t, \lambda_h^t)$ approaches the fixed point $(\hat{\mathbf{s}}^\infty, \lambda_\eta^\infty, \lambda_h^\infty)$, then $\hat{\mathbf{s}}^\infty$ is the solution to both Bayesian inference in (2) and optimal M-estimation in (5), with optimal loss and regularizer given by (18) with the choice of smoothing parameters $\lambda_\eta^\infty$ and $\lambda_h^\infty$.

We now discuss how to determine $\lambda_\eta^\infty$ and $\lambda_h^\infty$ analytically, thereby completing our heuristic derivation of an optimal M-estimator that matches Bayesian MMSE inference. An essential tool is state evolution (SE) which characterizes the gAMP dynamics [12] as follows. First, let $\mathbf{z} = \mathbf{X}\mathbf{s^0}$ be related to the true measurements. Then (6) implies that $\boldsymbol{\eta}^t - \mathbf{z}$ is a time-dependent residual. Remarkably, the gAMP equations ensure that the components of the residual $\boldsymbol{\eta}^t - \mathbf{z}$, as well as $\mathbf{h}^t = -\lambda_h^t \mathbf{X}^T G_y(\lambda_\eta^t, \mathbf{y}, \boldsymbol{\eta}^t)$ are Gaussian distributed; the history term in the update of $\boldsymbol{\eta}^t$ in (6) crucially cancels out non-Gaussian structure that would otherwise develop as the vectors $\boldsymbol{\eta}^t$ and $\mathbf{h}^t$ propagate through the nonlinear measurement and signal correction steps induced by $G_y$ and $G_s$. We denote by $q_\eta^t$ and $q_h^t$ the variance of the components of $\boldsymbol{\eta}^t - \mathbf{z}$ and $\mathbf{h}^t$ respectively. Additionally, we denote by $q_s^t = \frac{1}{P}\langle \|\hat{\mathbf{s}}^t - \mathbf{s}^0\|^2 \rangle$ the per component MSE at iteration $t$. SE is a set of analytical evolution equations for the quantities $(q_s^t, q_\eta^t, q_h^t, \lambda_\eta^t, \lambda_h^t)$ that characterize the state of gAMP. A rigorous derivation both for dense [12] Gaussian measurements and sparse measurements [13] reveal that the SE equations accurately track the gAMP dynamical state in the high dimensional limit $N, P \to \infty$ with $\alpha = \frac{N}{P} \, O(1)$ that we consider here.

We derive the specific form of the mAMP SE equations, yielding a set of 5 update equations (see SM section 3.1 for further details). We also derive the SE equations for bAMP, which are simpler. First, we find the relations $\lambda_\eta^t = q_\eta^t$ and $\lambda_h^t = q_h^t$. Thus SE for bAMP reduces to a pair of update equations:

$$q_\eta^{t+1} = \gamma \left\langle \left( G_s^B(q_h^t, s^0 + \sqrt{q_h^t} w) - s^0 \right)^2 \right\rangle_{w, s^0} \qquad q_h^t = \left( \alpha \gamma \left\langle \left( G_y^B(q_\eta^t, y, \eta^t) \right)^2 \right\rangle_{y, z, \eta^t} \right)^{-1}.$$

(19)

Here $w$ is a zero mean unit variance Gaussian and $s^0$ is a scalar signal drawn from the signal distribution $P_s$. Thus the computation of the next residual $q_\eta^{t+1}$ on the LHS of (19) involves computing the MSE in estimating a signal $s^0$ corrupted by Gaussian noise of variance $q_h^t$, using MMSE inference as an estimation prcoedure via the function $G^B$ defined in (13). The RHS involves an average over the joint distribution of scalar versions of the output $y$, true measurement $z$, and estimated measurement $\eta^t$. These three scalars are the SE analogs of the gAMP variables $\mathbf{y}$, $\mathbf{z}$, and $\boldsymbol{\eta}^t$, and they model the joint distribution of single components of these vectors. Their joint distribution is given by $P(y, z, \eta^t) = P_{y|z}(y|z)P(z, \eta^t)$. In the special case of bAMP, $z$ and $\eta^t$ are jointly zero mean Gaussian with second moments given by $\langle (\eta^t)^2 \rangle = \gamma \sigma_s^2 - q_\eta^t$, $\langle z^2 \rangle = \gamma \sigma_s^2$,

and $\langle z\eta^t \rangle = \gamma\sigma_s^2 - q_\eta^t$ (see SM 3.2 for derivations). These moments imply the residual variance $\langle (z - \eta^t)^2 \rangle = q_\eta^t$. Intuitively, when gAMP works well, that is reflected in the SE equations by the reduction of the residual variance $q_\eta^t$ over time, as the time dependent estimated measurement $\eta^t$ converges to the true measurement $z$. The actual measurement outcome $y$, after the nonlinear part of the measurement process, is always conditionally independent of the estimated measurement $\eta^t$, given the true linear part of the measurement, $z$. Finally, the joint distribution of a single component of $\hat{\mathbf{s}}^{t+1}$ and $\mathbf{s}^0$ in gAMP are predicted by SE to have the same distribution as $\hat{s}^{t+1} = G_s^B(q_h^t, s^0 + \sqrt{q_h^t}w)$, after marginalizing out $w$. Comparing with the LHS of (19) then yields that the MSE per component satisfies $q_s^t = q_\eta^t/\gamma$.

Now, bAMP performance, upon convergence, is characterized by the fixed point of SE, which satisfies

$$q_s = \mathrm{MMSE}(s^0|s^0 + \sqrt{q_h}w) \qquad q_h = \frac{1}{\alpha\gamma J\left[\,P_y(y|\eta,\gamma q_s)\,\right]}. \qquad (20)$$

Here, the MMSE function denotes the minimal error in estimating the scalar signal $s^0$ from a measurement of $s^0$ corrupted by additive Gaussian noise of variance $q_h$ via computation of the posterior mean $\langle\, s^0|s^0 + \sqrt{q_h}w \,\rangle$:

$$\mathrm{MMSE}(s^0|s^0 + \sqrt{q_h}w) = \left\langle\, \left(\langle\, s^0|s^0 + \sqrt{q_h}w \,\rangle - s^0\right)^2 \,\right\rangle_{s^0,w}. \qquad (21)$$

Also, the function $J$ on the RHS of (20) denotes the average Fisher information that $y$ retains about an input, with some additional Gaussian input noise of variance $q$:

$$J\left[\,P_y(y|\eta,q)\,\right] = -\left\langle\, \frac{\partial^2}{\partial\eta^2}\log P_y(y|\eta,q) \,\right\rangle_{\eta,y} \qquad (22)$$

These equations characterize the performance of bAMP, through $q_s$. Furthermore, they yield the optimal smoothing parameters $\lambda_\eta = \gamma q_s$ and $\lambda_h = q_h$. This choice of smoothing parameters, when used in (18), yield a *fixed* optimal loss $\mathcal{L}^{\mathrm{opt}}$ and regularizer $\sigma^{\mathrm{opt}}$. When this optimal loss and regularizer are used in the M-estimation problem in (5), the resulting M-estimator should have performance equivalent to that of MMSE inference in (2). This completes our heuristic derivation of an equivalence between optimal M-estimation and Bayesian inference through message passing.

In Figure 2 we demonstrate numerically that the optimal M-estimator substantially outperforms MAP, especially at low measurement density $\alpha$, and has performance equivalent to MMSE inference, as theoretically predicted by SE for bAMP.

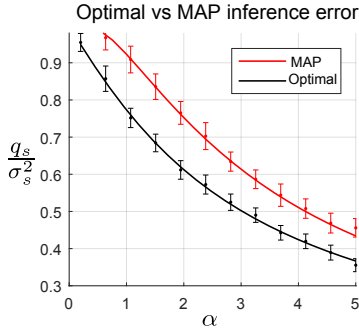

Figure 2: For logistic output and Laplacian signal, as in Fig. 1, we plot the per component MSE, normalized by signal variance. Smooth curves are theoretical predictions based on SE fixed points for mAMP for MAP inference (red) and bAMP for MMSE inference (black). Error bars reflect standard deviation in performance obtained by solving (5), via mAMP, for MAP inference (red) and optimal M-estimation (black), using simulated data generated as in (1), with dense i.i.d Gaussian measurements. For these finite simulated data sets, we varied $\alpha = \frac{N}{P}$, while holding $\sqrt{NP} \approx 250$. These results demonstrate that optimal M-estimation both significantly outperforms MAP (black below red) and matches Bayesian MMSE inference as predicted by SE for bAMP (black error bars consistent with black curve).

## 5   Discussion

Overall we have derived an optimal M-estimator, or a choice of optimal loss and regularizer, such the M-estimation problem in (5) has equivalent performance to that of Bayes optimal MMSE inference in (2), in the case of log-concave signal distribution and noise likelihood. Our derivation is heuristic in that it employs the formalism of gAMP, and as such depends on the correctness of a few statements. First, we assume that two special cases of the gAMP dynamics in (6), namely mAMP in (8) and

bAMP in (13) correctly solve the M-estimation problem in (5) and Bayesian MMSE inference in (2), respectively. We provide a heuristic derivation of both of these assumptions in the SM based on approximations of BP. Second, we require that SE in (19) correctly tracks the performance of gAMP in (13). We note that under mild conditions, the correctness of SE as a description of gAMP was rigorously proven in [12].

While we have not presented a rigorous derivation that the bAMP dynamics correctly solves the MMSE inference problem, we note several related rigorous results. First, it has been shown that bAMP is equivalent to MMSE inference in the limit of large sparse measurement matrices in [13, 14]. Also, in this same large sparse limit, the corresponding mAMP algorithm was shown to be equivalent to MAP inference with additive Gaussian noise [15]. In the setting of dense measurements, the correctness of bAMP has not yet been rigorously proven, but the associated SE is believed to be exact in the dense iid Gaussian measurement setting based on replica arguments from statistical physics (see e.g. section 4.3 in [16] for further discussion). For this reason, similar arguments have been used to determine theoretical bounds on inference algorithms in compressed sensing [16], and matrix factorization [17].

There are further rigorous results in the setting of M-estimation: mAMP and its associated SE is also provably correct in the large sparse measurement limit, and has additionally been rigorously proven to converge in special cases [5],[6] for dense iid Gaussian measurements. We further expect these results to generalize to a universality class of measurement matrices with iid elements and a suitable condition on their moments. Indeed this generalization was demonstrated rigorously for a subclass of M-estimators in [18]. In the setting of dense measurements, due to the current absence of rigorous results demonstrating the correctness of bAMP in solving MMSE inference, we have also provided numerical experiments in Fig. 2. This figure demonstrates that optimal M-estimation can significantly outperform MAP for high dimensional inference problems, again for the case of log-concave signal and noise.

Additionally, we note that the per-iteration time complexity of the gAMP algorithms (6, 7) scales linearly in both the number of measurements and signal dimensions. Therefore the optimal algorithms we describe are applicable to large-scale problems. Moreover, at lower measurement densities, the optimal loss and regularizer are smoother. Such smoothing may accelerate convergence time. Indeed smoother convex functions, with smaller Lipschitz constants on their derivative, can be minimized faster via gradient descent. It would be interesting to explore whether a similar result may hold for gAMP dynamics.

Another interesting future direction is the optimal estimation of sparse signals, which typically do not have log-concave distributions. One potential strategy in such scenarios would be to approximate the signal distribution with the best log-concave fit and apply optimal smoothing to determine a good regularizer. Alternatively, for any practical problem, one could choose the precise smoothing parameters through any model selection procedure, for example cross-validation on held-out data. Thus the combined Moreau and Gaussian smoothing in (18) could yield a family of optimization problems, where one member of this family could potentially yield better performance in practice on held-out data. For example, while LASSO performs very well for sparse signals, as demonstrated by its success in compressed sensing [19, 20], the popular elastic net [3], which sometimes outperforms pure LASSO by combining $L^1$ and $L^2$ penalties, resembles a specific type of smoothing of an $L^1$ regularizer. It would be interesting to see if combined Moreau and Gaussian smoothing underlying our optimal M-estimators could significantly out-perform LASSO and elastic net in practice, when our distributional assumptions about signal and noise need not precisely hold. However, finding optimal M-estimators for known sparse signal distributions, and characterizing the gap between their performance and that of MMSE inference, remains a fundamental open question.

## Acknowledgements

The authors would like to thank Lenka Zdeborova and Stephen Boyd for useful discussions and also Chris Stock and Ben Poole for comments on the manuscript. M.A. thanks the Stanford MBC and SGF for support. S.G. thanks the Burroughs Wellcome, Simons, Sloan, McKnight, and McDonnell foundations, and the Office of Naval Research for support.

## Footnotes

[1]Please see https://ganguli-gang.stanford.edu/pdf/16.Bayes.Mestimation.Supp.pdf for the supplementary material.

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
