[Supplementary Material]

# An equivalence between high dimensional Bayes optimal inference and M-estimation Supplementary Material

**Madhu Advani**     **Surya Ganguli**
Department of Applied Physics, Stanford University
msadvani@stanford.edu    and    sganguli@stanford.edu

## Contents

Here we provide additional derivations and examples to supplement the findings in the main paper.

# 1 Inference formulation

In this work we consider N samples $(\mathbf{x}_\mu, y_\mu)$ drawn from a generalized linear model:

$$y_\mu = r(\mathbf{x}_\mu \cdot \mathbf{s}^0, \epsilon_\mu), \tag{1}$$

where $\mathbf{s}^0 \in \mathbb{R}^P$ is a set of parameters to be inferred and $\epsilon_\mu$ denotes noise. We will be interested in the high dimensional limit of large numbers of both samples and parameters: $P, N \to \infty$, but with a finite measurement density $\alpha = \frac{N}{P} < \infty$. In particular we analyze and compare two methods for selecting $\hat{s}$, the parameter estimate: MMSE inference and regularized M-estimation.

## 1.1 MMSE inference

MMSE inference involves computing the $P$ dimensional integral:

$$\hat{s}_i^{\text{MMSE}} = \int s_i^0 P(\mathbf{s}^0 | \mathbf{X}, \mathbf{y}) d\mathbf{s}^0. \tag{2}$$

Here $\mathbf{X}$ denotes the measurement matrix where each row of the matrix is a measurement $\mathbf{x}_\mu$. MMSE inference minimizes the mean squared error $\left\langle (\hat{s}(\mathbf{X}, \mathbf{y}) - s^0)^2 \right\rangle$, a fact which can be seen by differentiating this expression with respect to $\hat{s}$ and setting the result equal to zero, which yields $\hat{s}(\mathbf{X}, \mathbf{y}) = \left\langle s^0 | \mathbf{X}, \mathbf{y} \right\rangle$ which is equivalent to (2). Here, $\left\langle \cdot \right\rangle$ denotes and average and $\left\langle x | y \right\rangle$ denotes the average of random variable $x$ given $y$.

To compute the MMSE estimate involves first computing the posterior distribution:

$$P(\mathbf{s}^0 | \mathbf{X}, \mathbf{y}) = \frac{P(\mathbf{y} | \mathbf{X}, \mathbf{s}^0) P(\mathbf{X}, \mathbf{s}^0)}{P(\mathbf{X}, \mathbf{y})} \tag{3}$$

We will assume throughout this work that the measurement matrix is chosen independently of the parameters $\mathbf{s}^0$ so that

$$P(\mathbf{s}^0 | \mathbf{X}, \mathbf{y}) \propto P(\mathbf{y} | \mathbf{X}, \mathbf{s}^0) P(\mathbf{s}^0). \tag{4}$$

Under the additional assumption of iid noisy channels and iid parameters, it follows that

$$P(\mathbf{s}^0 | \mathbf{X}, \mathbf{y}) \propto \prod_{\mu=1}^N P_{y|z}(y_\mu | \mathbf{x}_\mu \cdot \mathbf{s}^0) \prod_{j=1}^P P_s(s_j^0), \tag{5}$$

where $P_s$ is the distribution of the parameters, which for simplicity we assume to be zero mean and have variance $\sigma_s^2$, and $P_{y|z}$ is the noisy channel which outputs $y$. Note that neither of the noise nor the parameter distribution need be Gaussian. In general the integral above is intractable to compute since $x_\mu$ mixes parameters so that the posterior cannot be factorized. Due to the general difficulty of computing this integral, an often used surrogate is to use an optimization problem from a family of M-estimators.

## 1.2 M-estimation

The M-estimation problem takes the form:

$$\hat{\mathbf{s}} = \arg \min_{\mathbf{s}} \left[ \sum_{\mu=1}^N \mathcal{L}(y_\mu, \mathbf{x}_\mu \cdot \mathbf{s}) + \sum_i \sigma(s_i) \right], \tag{6}$$

where $\mathcal{L}(y, \eta)$ is convex (in $\eta$) loss function and $\sigma$ is a convex regularizer. Well known examples of such estimators which are commonly applied include [1] LASSO: $\mathcal{L}(y, \eta) = \frac{1}{2}(y - \eta)^2$ and $\sigma(s) = |s|$, Ridge regression: $\mathcal{L}(y, \eta) = \frac{1}{2}(y - \eta)^2$ and $\sigma(s) = \frac{1}{2}s^2$, and [2] Elastic Net: $\mathcal{L}(y, \eta) = \frac{1}{2}(y - \eta)^2$ and $\sigma(s) = \alpha|s| + \frac{\beta}{2}s^2$. We are interested in characterizing the performance of regularized M-estimators more generally and demonstrating that under the appropriate choice of $\mathcal{L}$ and $\sigma$, it is possible to achieve MMSE accuracy.

## 2 Approximate message passing

In this work we use Approximate Message Passing (AMP) primarily as a technique for deriving our main result about an equivalence between M-estimation and MMSE inference. Consider the following message passing algorithm which aims to compute the solutions to M-estimation optimization or MMSE inference:

$$\boldsymbol{\eta}^t = \mathbf{X}\hat{\mathbf{s}}^t + \lambda_\eta^t G_y(\lambda_\eta^{t-1}, \mathbf{y}, \boldsymbol{\eta}^{t-1}), \tag{7}$$

$$\lambda_h^t = \left( \frac{\gamma\alpha}{N} \sum_{\nu=1}^N \frac{\partial}{\partial\eta} G_y(\lambda_\eta^t, y_\nu, \eta_\nu) \right)^{-1}, \tag{8}$$

$$\hat{\mathbf{s}}^{t+1} = G_s\left( \lambda_h^t, \hat{\mathbf{s}}^t - \lambda_h^t \mathbf{X}^T G_y(\lambda_\eta^t, \mathbf{y}, \boldsymbol{\eta}^t) \right), \tag{9}$$

$$\lambda_\eta^{t+1} = \gamma\lambda_h^t \frac{1}{P} \sum_{j=1}^P \frac{\partial}{\partial h} G_s\left( \lambda_h^t, \hat{s}_j^t - \lambda_h^t \mathbf{X}^T G_y(\lambda_\eta^t, \mathbf{y}, \boldsymbol{\eta}^t) \right). \tag{10}$$

For the case of M-estimation, $G_y, G_s$ depend on the loss and regularization functions respectively and are defined as:

$$G_y(\lambda_\eta, y, \eta) = \mathcal{M}_{\lambda_\eta}[\mathcal{L}(y, \cdot)]'(\eta), \tag{11}$$

$$G_s(\lambda_h, h) = \mathcal{P}_{\lambda_h}[\sigma](h). \tag{12}$$

In the case of MMSE inference, we instead choose:

$$G_y(\lambda_\eta, y, \eta) = -\frac{\partial}{\partial\eta} \log\left( \int P_y(y|z) e^{-\frac{(\eta-z)^2}{2\lambda_\eta}} dz \right), \tag{13}$$

$$G_s(\lambda_h, h) = h + \lambda_h \frac{\partial}{\partial h} \log\left( \int P_s(s) e^{-\frac{(h-s)^2}{2\lambda_h}} ds \right). \tag{14}$$

We will provide a heuristic derivation in sections 2.2 and 2.3 of this message passing algorithm and the form of $G_y, G_s$ based on an analytic relaxation of loopy belief propagation for bAMP and mAMP respectively. We include this derivation since since it illustrates that the approximations we make are valid in the limit of large sparse measurement matrices. However, the AMP algorithm and predictions we derive about its performance are also exact for larger class of non-sparse measurement matrices, as we demonstrate with simulations in the main text. There are rigorous derivations based on AMP (see [3],[4]) which prove rigorously that special cases of this algorithm converge on loopy graphs. While we do not aim to provide a rigorous proof of the convergence of AMP in this work, we perform numerical simulations in the main paper and in the following section we show that the fixed points of the AMP algorithm are solutions to the M-estimation optimization problem

### 2.1 Fixed points of mAMP are minima of M-estimation

Under the choice (11, 12), the mAMP algorithm has the form:

$$\boldsymbol{\eta}^t = \mathbf{X}\hat{\mathbf{s}}^t + \lambda_\eta^t \mathcal{M}_{\lambda_\eta^{t-1}}[\mathcal{L}(\mathbf{y}, \cdot)]'(\boldsymbol{\eta}^{t-1}), \tag{15}$$

$$\hat{\mathbf{s}}^{t+1} = \mathcal{P}_{\lambda_h^t}[\sigma](\hat{\mathbf{s}}^t - \lambda_h^t \mathbf{X}^T \mathcal{M}_{\lambda_\eta^t}[\mathcal{L}(\mathbf{y}, \cdot)]'(\boldsymbol{\eta}^t)). \tag{16}$$

We now show that fixed points of the AMP algorithm are critical points of the M-estimator optimization problem for mAMP . We consider a fixed point of (15) by dropping the $t$ index and rearranging the expression to yield:

$$\boldsymbol{\eta} - X\hat{\mathbf{s}} = \lambda_\eta \mathcal{M}_{\lambda_\eta}[\mathcal{L}(\mathbf{y}, \cdot)]'(\boldsymbol{\eta}) = \boldsymbol{\eta} - \mathcal{P}_{\lambda_\eta}[\mathcal{L}(\mathbf{y}, \cdot)](\boldsymbol{\eta}), \tag{17}$$

where the final equality follows from the fact that the proximal map can be understood as a gradient descent step along the Moreau envelope, see appendix A.1. We will also use the fact that the proximal map is related to the derivative of the function it maps via the equation

$$x - \mathcal{P}_\lambda[f](x) = \lambda f'(\mathcal{P}_\lambda[f](x)). \tag{18}$$

For a derivation of this fact, see appendix A.2. If follows that

$$\boldsymbol{\eta} - \mathcal{P}_{\lambda_\eta}[\mathcal{L}(\mathbf{y}, \cdot)](\boldsymbol{\eta}) = \lambda_\eta \frac{\partial}{\partial \eta} \mathcal{L}(y, \mathcal{P}_{\lambda_\eta}[\mathcal{L}(\mathbf{y}, \cdot)](\boldsymbol{\eta})). \tag{19}$$

Combining (17) with (19), the two preceding equations yield:

$$\mathcal{M}_{\lambda_\eta}[\mathcal{L}(\mathbf{y}, \cdot)]'(\boldsymbol{\eta}) = \frac{\partial}{\partial \eta} \mathcal{L}(y, \mathbf{X}\hat{\mathbf{s}}). \tag{20}$$

If we now define $\mathbf{h} = \hat{\mathbf{s}} - \lambda_h \mathbf{X}^T \mathcal{M}_{\lambda_\eta}[\mathcal{L}(\mathbf{y}, \cdot)]'(\boldsymbol{\eta})$, then it follows that

$$\lambda_h \sigma'(\hat{\mathbf{s}}) = \lambda_h \sigma'(\mathcal{P}_{\lambda_h}[\sigma](\mathbf{h})) = \mathbf{h} - \hat{\mathbf{s}} = -\lambda_h \mathbf{X}^T \mathcal{M}_{\lambda_\eta}[\mathcal{L}(\mathbf{y}, \cdot)]'(\boldsymbol{\eta}) = -\lambda_h \mathbf{X}^T \frac{\partial}{\partial \eta} \mathcal{L}(y, \mathbf{X}\hat{\mathbf{s}}), \tag{21}$$

where the second equality follows from (18) and the final equality follows from (20). Dividing both sides of the equality above by $\lambda_h$ and rearranging, the fixed points of AMP satisfy:

$$\mathbf{X}^T \partial_\eta \mathcal{L}(\mathbf{y}, \mathbf{X}\hat{\mathbf{s}}) + \sigma'(\hat{\mathbf{s}}) = \mathbf{0}, \tag{22}$$

and must be fixed points of the M-estimation optimization (6).

## 2.2 Derivation of bAMP

$$\hat{s}_i^{\mathrm{mmse}} = \int s_i P(\mathbf{s}|\mathbf{X}, \mathbf{y}) d\mathbf{s}. \tag{23}$$

The belief propagation (BP) [5] equations for estimating the marginal distribution for each parameter $s_i$ are simply:

$$m_{i \to \mu}^{t+1}(s_i) = P_s(s_i) \prod_{\nu \neq \mu} m_{\nu \to i}^t(s_i), \tag{24}$$

$$m_{\mu \to i}^t(s_i) = \int P_y(y_\mu | \sum_j X_{\mu j} s_j) \prod_{j \neq i} m_{j \to \mu}^t(s_j) \prod_{j \neq i} ds_j. \tag{25}$$

The BP equations above rapidly converge to the true marginal distributions if the corresponding factor graph is a tree so that it contains no loops. We will also be interested in dense graphs, see the discussion in the main paper for why AMP is relevant in this setting. The factor graph can be understood as a set of factor nodes corresponding to each sample $\mu$ and variable nodes corresponding to each parameter $i$, for more discussion on this see [6]. We follow the same derivation technique as this work and begin by approximating messages from parameters to factors as an exponential of a quadratic. The approximation is justified because if more terms were included as a Taylor series in the exponent, third and higher order terms would have a negligible effect for a large class of measurement matrices which have sufficiently sparse or random elements so that $\sum_j X_{\mu j}^3 \to 0$ in the asymptotic limit. We will also assume that the measurement elements are iid and scaled so that the average squared norm of the measurements satisfy: $\langle \mathbf{x}_\mu \cdot \mathbf{x}_\mu \rangle = \gamma$. Under a second order Taylor expansion in $s_j$, the messages from parameters to factors may be written as

$$m_{j \to \mu}^t(s_j) \approx \frac{1}{\sqrt{2\pi \lambda_{j \to \mu}^t}} e^{-\frac{(s_j - \hat{s}_{j \to \mu}^t)^2}{2\lambda_{j \to \mu}^t}}, \tag{26}$$

which upon substitution into (25) yields:

$$m_{\mu \to i}^t(s_i) \cong \int P_y(y_\mu | \sum_j X_{\mu j} s_j) \prod_{j \neq i} \left( \frac{1}{\sqrt{2\pi \lambda_{j \to \mu}^t}} e^{-\frac{(s_j - \hat{s}_{j \to \mu}^t)^2}{2\lambda_{j \to \mu}^t}} ds_j \right). \tag{27}$$

Under the change of variables $r_j = \frac{s_j}{\sqrt{\lambda_{j \to \mu}^t}}, \hat{r}_{j \to \mu} = \frac{\hat{s}_{j \to \mu}}{\sqrt{\lambda_{j \to \mu}^t}}$, the integral above simplifies to

$$\int P_y(y_\mu | \sum_j \sqrt{\lambda_{j\to\mu}^t} X_{\mu j} r_j) \prod_{j\neq i} \left( \frac{1}{\sqrt{2\pi}} e^{-\frac{(r_j - \hat{r}_{j\to\mu}^t)^2}{2}} dr_j \right). \tag{28}$$

The above integral may be reduced to an single dimensional integral since $\sum_{j\neq i} \sqrt{\lambda_{j\to\mu}^t} X_{\mu j} r_j$ is

constant under changes in $\mathbf{r}$ orthogonal to $\boldsymbol{\chi}^\mu = \sum_j \frac{X_{\mu j} \sqrt{\lambda_{j\to\mu}^t}}{\left( \sum_{j\neq i} X_{\mu j}^2 \lambda_{j\to\mu}^t \right)^{1/2}} \hat{e}_j$, also the dependence of

the denominator on $\mu$ and $i$ becomes weak in the asymptotic limit so that it approaches a scalar:
$\lambda_\eta^t = \sum_{j\neq i} X_{\mu j}^2 \lambda_{j\to\mu}^t$. The one dimensional integral becomes:

$$m_{\mu\to i}^t(s_i) \cong \int P_y(y_\mu | \sum_{j\neq i} \sqrt{\lambda_{j\to\mu}^t} X_{\mu j} \hat{r}_{j\to\mu}^t + \sqrt{\lambda_\eta^t} v + X_{\mu i} s_i) \frac{1}{\sqrt{2\pi}} e^{-\frac{v^2}{2}} dv, \tag{29}$$

where we have defined $r_j = \hat{r}_{j\to\mu}^t + v\chi_j^\mu$. Under the further change of variables $\xi = \sum_{j\neq i} \sqrt{\lambda_{j\to\mu}^t} X_{\mu j} \hat{r}_{j\to\mu}^t + \sqrt{\lambda_\eta^t} v + X_{\mu i} s_i$, the integral is unchanged:

$$m_{\mu\to i}^t(s_i) \cong \int P_y(y_\mu | \xi) \frac{1}{\sqrt{2\pi\lambda_\eta^t}} e^{-\frac{1}{2\lambda_\eta^t}(\xi - X_{\mu i} s_i - \sum_{j\neq i} X_{\mu j} \hat{s}_{j\to\mu}^t)^2} d\xi. \tag{30}$$

We then again make a Gaussian approximation for the set of marginals from factors to parameters:

$$m_{\mu\to i}^t(s_i) \cong e^{\alpha_{\mu\to i}^t X_{\mu i} s_i - \frac{1}{2} \beta_{\mu\to i}^t X_{\mu i}^2 s_i^2}. \tag{31}$$

It follows from differentiating (30) and (31) with respect to $s_i$ and solving for $\alpha_{\mu\to i}^t$ that

$$\alpha_{\mu\to i}^t = \frac{\frac{1}{\lambda_\eta^t} \int P_y(y_\mu | \xi) \left( \xi - \sum_{j\neq i} X_{\mu j} \hat{s}_{j\to\mu}^t \right) \frac{1}{\sqrt{2\pi\lambda_\eta^t}} e^{-\frac{1}{2\lambda_\eta^t}(y_\mu - \sum_{j\neq i} X_{\mu j} \hat{s}_{j\to\mu}^t - \xi)^2} d\xi}{\int P_y(y_\mu | \xi) \frac{1}{\sqrt{2\pi\lambda_\eta^t}} e^{-\frac{1}{2\lambda_\eta^t}(y_\mu - \sum_{j\neq i} X_{\mu j} \hat{s}_{j\to\mu}^t - \xi)^2} d\xi}, \tag{32}$$

which we write in the form:

$$\alpha_{\mu\to i}^t = -G_y^B(\lambda_\eta^t, y_\mu, \sum_{j\neq i} X_{\mu j} \hat{s}_{j\to\mu}^t), \tag{33}$$

where

$$G_y^B(\lambda, y, \eta) = \frac{\frac{1}{\lambda} \int P_y(y|z)(\eta - z) \frac{1}{\sqrt{2\pi\lambda}} e^{-\frac{(\eta-z)^2}{2\lambda}} dz}{\int P_y(y|z) \frac{1}{\sqrt{2\pi\lambda}} e^{-\frac{(\eta-z)^2}{2\lambda}} dz}. \tag{34}$$

It also follows from (30) and (31) that

$$\beta_{\mu\to i}^t = \frac{\partial}{\partial\eta} G_y^B(\lambda_\eta^t, y_\mu, \sum_{j\neq i} X_{\mu j} \hat{s}_{j\to\mu}^t). \tag{35}$$

Substituting the approximate form of messages from factors to parameters (31) into the belief propagation update equation (24) yields

$$m_{i\to\mu}^{t+1}(s_i) = P_s(s_i) e^{\sum_{\nu\neq\mu} \alpha_{\nu\to i}^t X_{\nu i} s_i - \frac{1}{2} \sum_{\nu\neq\mu} \beta_{\nu\to i}^t X_{\nu i}^2 s_i^2}. \tag{36}$$

Defining $\frac{1}{\lambda_h^t} = \sum_\nu X_{\nu i}^2 \beta_{\nu\to i}^t$, in the asymptotic limit yields

$$m_{i\to\mu}^{t+1}(s_i) \cong P_s(s_i) e^{\sum_{\nu\neq\mu} \alpha_{\nu\to i}^t X_{\nu i} s_i - \frac{1}{2\lambda_h^t} s_i^2} \cong P_s(s_i) e^{-\frac{1}{2\lambda_h^t}(s_i - \lambda_h^t \sum_{\nu\neq\mu} \alpha_{\nu\to i}^t X_{\nu i})^2}. \tag{37}$$

It also follows from the definition of $\lambda_h^t$ that

$$\lambda_h^t = \left( \sum_{\nu=1}^N X_{\nu i}^2 \frac{\partial}{\partial\eta} G_y^B(\lambda_\eta^t, y_\nu, \sum_{j\neq i} X_{\nu j} \hat{s}_{j\to\nu}^t) \right)^{-1} \to \left( \frac{\alpha\gamma}{N} \sum_{\nu=1}^N \frac{\partial}{\partial\eta} G_y^B(\lambda_\eta^t, y_\nu, \sum_{j\neq i} X_{\nu j} \hat{s}_{j\to\nu}^t) \right)^{-1}. \tag{38}$$

The arrow denotes the limit of $\lambda_h^t$ for very large system sizes. It follows from our quadratic approximation (26) of the marginals, that the mean of the marginal satisfies

$$\hat{s}_{i \to \mu}^t = \frac{\int s_i m_{i \to \mu}^t(s_i) ds_i}{\int m_{i \to \mu}^t(s_i) ds_i}, \tag{39}$$

and that the variance satisfies

$$\lambda_{i \to \mu}^t = \frac{\int (s_i - s_{i \to \mu}^t)^2 m_{i \to \mu}^t(s_i) ds_i}{\int m_{i \to \mu}^t(s_i) ds_i}. \tag{40}$$

If we define

$$G_s^B(\lambda, h) = \frac{\int s P_s(s) e^{-\frac{1}{2\lambda}(s-h)^2} ds}{\int P_s(s) e^{-\frac{1}{2\lambda}(s-h)^2} ds}, \tag{41}$$

then

$$\hat{s}_{i \to \mu}^{t+1} = G_s^B(\lambda_h^t, \sum_{\nu \neq \mu} \lambda_h^t \alpha_{\nu \to i}^t X_{\nu i}), \tag{42}$$

$$\lambda_{i \to \mu}^{t+1} = \lambda_h^t \frac{\partial}{\partial h} G_s^B(\lambda_h^t, \sum_{\nu \neq \mu} \lambda_h^t \alpha_{\nu \to i}^t X_{\nu i}). \tag{43}$$

It then follows from our definition of $\lambda_\eta^t$ that

$$\lambda_\eta^{t+1} = \sum_{i=1}^P X_{\mu i}^2 \lambda_{i \to \mu}^{t+1} \to \gamma \frac{1}{P} \sum_{i=1}^P \lambda_{i \to \mu}^{t+1}, \tag{44}$$

thus

$$\lambda_\eta^{t+1} = \gamma \lambda_h^t \frac{1}{P} \sum_{i=1}^P \frac{\partial}{\partial h} G_s^B(\lambda_h^t, \sum_{\nu \neq \mu} \lambda_h^t \alpha_{\nu \to i}^t X_{\nu i}). \tag{45}$$

We can equivalently write $G_s^B, G_y^B$ in the forms:

$$G_s^B(\lambda, h) = h + \lambda \frac{d}{dh} \log \left( \int P_s(s) e^{-\frac{(h-s)^2}{2\lambda}} ds \right), \tag{46}$$

$$G_y^B(y, \lambda, \eta) = -\frac{d}{d\eta} \log \left( \int P_y(y|z) e^{-\frac{(\eta-z)^2}{2\lambda}} dz \right). \tag{47}$$

This form will be useful later when we discuss the connection between bAMP and mAMP.

## 2.3  Derivation of mAMP

In this section we derive an AMP algorithm to solve the M-estimation optimization problem

$$\hat{s} = \arg \min_s \left[ \sum_{\mu=1}^N \mathcal{L}(y_\mu, \mathbf{x}_\mu \cdot \mathbf{s}) + \sum_{i=1}^P \sigma(s_i) \right]. \tag{48}$$

This optimization also admits a factor graph and the derivation of BP on this factor graph is very similar to that of the previous section. The algorithm is intended to solve a minimization problem and is referred to in the literature as the min-sum algorithm [6]. Again the result is guaranteed to converge to the correct solution on a tree-like factor graph, but remarkably there are rigorous results demonstrating that the algorithm is correct even on dense graphs in special cases (e.g. LASSO in [3]). The min-sum BP algorithm used to solve M-estimation takes the form:

$$\hat{J}_{\mu \to i}^t(s_i) = \min_{s_{\partial \mu \setminus i}} \left[ \mathcal{L}(y_\mu, \mathbf{x}_\mu \cdot \mathbf{s}) + \sum_{j \in \partial \mu \setminus i} J_{j \to \mu}^t(s_j) \right], \tag{49}$$

$$J_{i \to \mu}^t(s_i) = \sigma(s_i) + \sum_{\nu \neq \mu} \hat{J}_{\nu \to i}^{t-1}(s_i). \tag{50}$$

After BP has run to convergence (denoted here by $t = \infty$), the estimates can be computed by a simple single dimensional minimization problem:

$$\hat{s}_i = \arg\min_{s_i} \left[ \sum_{\nu=1}^{N} \hat{J}_{\nu \to i}^{\infty}(s_i) + \sigma(s_i) \right]. \tag{51}$$

To avoid the computational cost of estimating the full functions $J_{i \to \mu}^t(s_i)$ (called a message) as in [6], we approximate them as a quadratic with a minimum at $\hat{s}_{i \to \mu}$, which may be thought of as the estimated parameters given that factor $\mu$ is removed from the graph. The message can be approximated with a Taylor expansion around its minima as:

$$J_{i \to \mu}^t(s_i) \cong \frac{1}{2\lambda_{i \to \mu}^t}(s_i - \hat{s}_{i \to \mu}^t)^2 + O\left((s_i - \hat{s}_{i \to \mu}^t)^3\right). \tag{52}$$

By substituting the form of this message into the min-sum equations (49,50), we can write message passing equations as:

$$\hat{J}_{\mu \to i}^t(s_i) = \min_{s_{\partial \mu \backslash i}} \left[ \mathcal{L}(y_\mu, \mathbf{x}_\mu \cdot \mathbf{s}) + \sum_{j \neq i} \frac{1}{2\lambda_{j \to \mu}^t}(s_j - s_{j \to \mu}^t)^2 + O\left(\sum_j (s_j - s_{j \to \mu}^t)^3\right) \right]. \tag{53}$$

Under the change of variables $w_j = \frac{s_j}{\sqrt{\lambda_{j \to \mu}}}$ and $w_{j \to \mu}^t = \frac{\hat{s}_{j \to \mu}^t}{\sqrt{\lambda_{j \to \mu}}}$, this minimization may be written as

$$\hat{J}_{\mu \to i}^t(s_i) = \min_{w_{\partial \mu \backslash i}} \left[ \mathcal{L}(y_\mu, \sum_{j \neq i} X_{\mu j} w_j \sqrt{\lambda_{j \to \mu}} + X_{\mu i} s_i) + \sum_{j \in \partial \mu \backslash i} \frac{1}{2}(w_j - w_{j \to \mu}^t)^2 + O\left(\sum_j (w_j - w_{j \to \mu}^t)^3\right) \right]. \tag{54}$$

The $\mathbf{w}_{\delta \mu \backslash i}$ which minimizes the expression above must have the form

$$w_j = w_{j \to \mu}^t + r X_{\mu j} \sqrt{\lambda_{j \to \mu}}, \tag{55}$$

where $r$ is a scalar. Substituting this expression into the equation for $\hat{J}_{\mu \to i}^t(s_i)$ yields:

$$\hat{J}_{\mu \to i}^t(s_i) = \min_r \left[ \mathcal{L}(y_\mu, \sum_{j \neq i} X_{\mu j} w_{j \to \mu}^t \sqrt{\lambda_{j \to \mu}} + r \sum_{j \neq i} X_{\mu j}^2 \lambda_{j \to \mu} + X_{\mu i} s_i) + \frac{r^2}{2} \sum_{j \neq i} X_{\mu j}^2 \lambda_{j \to \mu} \right]. \tag{56}$$

The higher order terms will be negligible in the large system limit under the assumption $\sum_j X_{\mu j}^3 \to 0$. We can removing $\mathbf{w}^t$ by writing the above expression in terms of $\hat{s}^t$ and can define $\lambda_\eta^t = \sum_j X_{\mu j}^2 \lambda_{j \to \mu}^t$, which simplifies the form of the message to a single variable minimization:

$$\hat{J}_{\mu \to i}^t(s_i) = \min_r \left[ \mathcal{L}(y_\mu, \sum_{j \backslash i} X_{\mu j} s_{j \to \mu}^t + r \lambda_\eta + X_{\mu i} s_i) + \lambda_\eta \frac{r^2}{2} \right]. \tag{57}$$

We can express the previous equation as a Moreau envelope, to do so we make the change of variables:

$$\xi = \sum_{j \backslash i} X_{\mu j} s_{j \to \mu}^t + r \lambda_\eta + X_{\mu i} s_i, \tag{58}$$

so that

$$\hat{J}_{\mu \to i}^t(s_i) = \min_\xi \left[ \mathcal{L}(y_\mu, \xi) + \frac{\left(\xi - \sum_{j \backslash i} X_{\mu j} s_{j \to \mu}^t - X_{\mu i} s_i\right)^2}{2\lambda_\eta} \right] = \mathcal{M}_{\lambda_\eta}[\mathcal{L}(y_\mu, \cdot)](X_{\mu i} s_i + \sum_{j \backslash i} X_{\mu j} s_{j \to \mu}^t). \tag{59}$$

The RHS of the equation above follows directly from the definition of the Moreau envelope. See appendix A for the definition of the Moreau envelope and some of its properties. The other form of message $\hat{J}_{\mu \to i}^t(s_i)$ can also be Taylor expanded in $s_i$ to yield

$$\hat{J}_{\mu \to i}^t(s_i) \cong X_{\mu i} s_i \mathcal{M}_{\lambda_\eta^t}[\mathcal{L}(y_\mu, \cdot)]'(\sum_{j \backslash i} X_{\mu j} s_{j \to \mu}^t) + \frac{X_{\mu i}^2 s_i^2}{2} \mathcal{M}_{\lambda_\eta^t}[\mathcal{L}(y_\mu, \cdot)]''(\sum_{j \neq i} X_{\mu j} s_{j \to \mu}^t) + O(X_{\mu i}^3 s_i^3). \tag{60}$$

When these messages are summed over $\mu$, the final term will be negligible in the large system limit, so we can write the above equation in the form:

$$\hat{J}^t_{\mu \to i}(s_i) \cong -\alpha^t_{\mu \to i} X_{\mu i} s_i + \frac{1}{2} \beta^t_{\mu \to i} X^2_{\mu i} s^2_i, \tag{61}$$

where

$$\alpha^t_{\mu \to i} = -\mathcal{M}_{\lambda^t_\eta}[\mathcal{L}(y_\mu, \cdot)]'(\textstyle\sum_{j \backslash i} X_{\mu j} s^t_{j \to \mu}). \tag{62}$$

Substituting the form (61) into (50) yields

$$J^t_{i \to \mu}(s_i) \cong \sigma(s_i) + \left( \sum_{\nu \neq \mu} -\alpha^{t-1}_{\nu \to i} X_{\nu i} \right) s_i + \frac{1}{2} \left( \sum_{\nu \neq \mu} \beta^{t-1}_{\nu \to i} X^2_{\nu i} \right) s^2_i. \tag{63}$$

We then define:

$$\lambda^t_h = \frac{1}{\sum_\nu \beta^t_{\nu \to i} X^2_{\nu i}}, \tag{64}$$

and remove the index $i$ because in the large system limit this parameter should converge to the same value for all $i$ by the law of large numbers. It then follows from the definition of $\beta^t_{\nu \to i}$ and $\hat{J}^t_{\mu \to i}(s_i)$ in (63, 61) that

$$\lambda^t_h = \left( \sum_{\nu=1}^N X^2_{\nu i} \mathcal{M}_{\lambda^t_\eta}[\mathcal{L}(y_\nu, \cdot)]''(\textstyle\sum_{j \neq i} X_{\nu j} \hat{s}^t_{j \to \nu}) \right)^{-1} \to \left( \frac{\alpha \gamma}{N} \sum_{\nu=1}^N \mathcal{M}_{\lambda^t_\eta}[\mathcal{L}(y_\nu, \cdot)]''(\textstyle\sum_{j \neq i} X_{\nu j} \hat{s}^t_{j \to \nu}) \right)^{-1}. \tag{65}$$

By definition, $\hat{s}^t_{i \to \mu}$ is the arg min of $J^t_{i \to \mu}(s_i)$, which implies that

$$\hat{s}^t_{i \to \mu} = \mathcal{P}_{\lambda^{t-1}_h}[\sigma](\lambda^{t-1}_h \textstyle\sum_{\nu \neq \mu} \alpha^{t-1}_{\nu \to i} X_{\nu i}), \tag{66}$$

where $\mathcal{P}_\lambda[\sigma](\cdot)$ is a proximal map as defined in appendix A. Using the form of $\lambda^t_\eta = \sum_j X^2_{ij} \lambda^t_{j \to \mu}$, and earlier assumption about the form of $\lambda$: $\lambda^t_{j \to \mu} = J''_{j \to \mu}(\hat{s}^t_{i \to \mu})$ we differentiate (63) twice and substitute the form of $\hat{s}^t_{i \to \mu}$ in (66) and apply a relation between the derivative of a function and a proximal map derived in appendix A.2, which yields:

$$\lambda^{t+1}_\eta = \lambda^t_h \sum_{i=1}^P X^2_{\mu i} \mathcal{P}_{\lambda^t_h}[\sigma]'(\textstyle\sum_{\nu \neq \mu} \lambda^t_h \alpha^t_{\nu \to i} X_{\nu i}) \to \gamma \lambda^t_h \frac{1}{P} \sum_{i=1}^P \mathcal{P}_{\lambda^t_h}[\sigma]'(\textstyle\sum_{\nu \neq \mu} \lambda^t_h \alpha^t_{\nu \to i} X_{\nu i}). \tag{67}$$

## 2.4 Simplification of AMP to require $O(N + P)$ messages

In this section we will simplify the AMP algorithms derived in the previous sections (2.2,2.3) so that rather than keeping track of O(NP) variables we can keep track of O(N+P) variables. Before doing so we note that the expressions derived in section 2.2 and 2.3 have the same form. If we define functions $G_s, G_y$ and choose for mAMP :

$$G_s(\lambda_h, h) = \mathcal{P}_{\lambda_h}[\sigma](h), \qquad G_y(\lambda_\eta, y, \eta) = \mathcal{M}_{\lambda_\eta}[\mathcal{L}(y, \cdot)]'(\eta), \tag{68}$$

and for bAMP :

$$G_s(\lambda_h, h) = h + \lambda_h \frac{\partial}{\partial h} \log(P_s(h, \lambda_h)), \qquad G_y(\lambda_\eta, y, \eta) = -\frac{\partial}{\partial \eta} \log(P_y(y|\eta, \lambda_\eta)), \tag{69}$$

then it is easy to check that the algorithms derived in 2.2 and 2.3 both have the form:

$$\hat{s}^{t+1}_{i \to \mu} = G_s(\lambda^t_h, \sum_{\nu \neq \mu} \lambda^t_h \alpha^t_{\nu \to i} X_{\nu i}), \tag{70}$$

$$\alpha^t_{\mu \to i} = -G_y(\lambda^t_\eta, y_\mu, \sum_{j \neq i} X_{\mu j} \hat{s}^t_{j \to \mu}), \tag{71}$$

where $\lambda_\eta, \lambda_h$ are updated via

$$\lambda_\eta^{t+1} = \gamma \lambda_h^t \frac{1}{P} \sum_{i=1}^{P} \frac{\partial}{\partial h} G_s(\lambda_h^t, \sum_{\nu \neq \mu} \lambda_h^t \alpha_{\nu \to i}^t X_{\nu i}), \tag{72}$$

$$\lambda_h^t = \left( \alpha \gamma \frac{1}{N} \sum_{\nu=1}^{N} \frac{\partial}{\partial \eta} G_y(\lambda_\eta^t, y_\nu, \sum_{j \neq i} X_{\nu j} \hat{s}_{j \to \nu}^t) \right)^{-1}. \tag{73}$$

It is possible to simplify the algorithm further and keep track of fewer parameters by Taylor expanding equations (70,71) in small quantities so that in the asymptotic limit, only the first order expansion is needed. To do this we look for solutions of the form $\hat{s}_{i \to \mu}^t = \hat{s}^t + \delta s_{i \to \mu}$ and $\alpha_{\mu \to i}^t = \hat{\alpha} + \delta \alpha_{\mu \to i}$ so that we can expand the update equations as

$$\hat{s}_i^{t+1} + \delta s_{i \to \mu}^{t+1} = G_s(\lambda_h^t, \sum_\nu \lambda_h^t \alpha_{\nu \to i}^t X_{\nu i} - \lambda_h^t \alpha_{\mu \to i}^t X_{\mu i}), \tag{74}$$

$$\alpha_\mu^t + \delta \alpha_{\mu \to i}^t = -G_y(\lambda_\eta^t, y_\mu, \sum_j X_{\mu j} \hat{s}_{j \to \mu}^t - X_{\mu i} \hat{s}_{i \to \mu}^t). \tag{75}$$

In the large system limit the $\delta \alpha$ and $\delta s$ terms will both be small as will the individual elements of $X$. Multiplying the two gives a result which is small squared which is the intuition for ignoring these terms in the derivation which follows. We can expand in the small terms $\lambda_h^t \alpha_{\mu \to i}^t X_{\mu i} \approx \lambda_h^t \alpha_\mu^t X_{\mu i}$ and $X_{\mu i} \hat{s}_{i \to \mu}^t \approx X_{\mu i} \hat{s}_i^t$. Matching indices in the resulting Taylor expansion yields expressions for $\delta s_{i \to \mu}^t, \delta \alpha_{\mu \to i}^t$:

$$\delta s_{i \to \mu}^{t+1} = -\partial_s G_s(\lambda_h^t, \sum_\nu \lambda_h^t \alpha_{\nu \to i}^t X_{\nu i}) \lambda_h^t \alpha_\mu^t X_{\mu i}, \tag{76}$$

$$\delta \alpha_{\mu \to i}^t = \partial_\eta G_y(\lambda_\eta^t, y_\mu, \sum_j X_{\mu j} \hat{s}_{j \to \mu}^t) X_{\mu i} \hat{s}_i^t. \tag{77}$$

Similarly,

$$\hat{s}_i^{t+1} = G_s(\lambda_h^t, \sum_\nu \lambda_h^t \alpha_{\nu \to i}^t X_{\nu i}) = G_s(\lambda_h^t, \sum_\nu \lambda_h^t (\alpha_\nu^t + \delta \alpha_{\nu \to i}^t) X_{\nu i}), \tag{78}$$

$$\alpha_\mu^t = -G_y(\lambda_\eta^t, y_\mu, \sum_j X_{\mu j} \hat{s}_{j \to \mu}^t) = -G_y(\lambda_\eta^t, y_\mu, \sum_j X_{\mu j} (\hat{s}_j^t + \delta s_{j \to \mu}^t)). \tag{79}$$

Thus, we can write the preceding equations as

$$\hat{s}_i^{t+1} = G_s(\lambda_h^t, h_i^t), \tag{80}$$

$$\alpha_\mu^t = -G_y(\lambda_\eta^t, y_\mu, \eta_\mu^t), \tag{81}$$

under the definition:

$$h_i^t = \sum_\nu \lambda_h^t (\alpha_\nu^t + \delta \alpha_{\nu \to i}^t) X_{\nu i}, \tag{82}$$

$$\eta_\mu^t = \sum_j X_{\mu j} (\hat{s}_j^t + \delta s_{j \to \mu}^t). \tag{83}$$

Substituting the form of $\delta s_{i \to \mu}$ from (76) allows us to write the update equation (83) as

$$\eta_\mu^t = \sum_j X_{\mu j} \hat{s}_j^t - \alpha_\mu^t \lambda_h^t \sum_j X_{\mu j}^2 \frac{\partial}{\partial h} G_s(\lambda_h^t, h_j^t). \tag{84}$$

We can similarly expand $h_i^t$ in (82) as

$$h_i^t = -\lambda_h^t \sum_\nu X_{\nu i} G_y(\lambda_\eta^t, y_\nu, \eta_\nu^t) + \hat{s}_i^t \lambda_h^t \sum_\nu X_{\nu i}^2 \frac{\partial}{\partial \eta} G_y(\lambda_\eta^t, y_\nu, \eta_\nu^t) = -\lambda_h^t \sum_\nu X_{\nu i} G_y(\lambda_\eta^t, y_\nu, \eta_\nu^t) + \hat{s}_i^t,$$
$$\tag{85}$$

where the final equality follows from the form of $\lambda_h$ (e.g. (65)). The final message passing equations are simply

$$\hat{s}_i^{t+1} = G_s(\lambda_h^t, \hat{s}_i^t - \lambda_h^t \sum_\nu X_{\nu i} G_y(\lambda_\eta^t, y_\nu, \eta_\nu^t)), \tag{86}$$

$$\eta_\mu^t = \sum_j X_{\mu j} \hat{s}_j^t + \lambda_\eta^t G_y(\lambda_\eta^{t-1}, y_\mu, \eta_\mu^{t-1}). \tag{87}$$

## 3 State evolution for AMP: theoretical predictions of algorithm performance

### 3.1 General state evolution relations

We can track the AMP algorithm performance at each iteration via a formalism called state evolution (SE), which allows one to derive a scalar characterization of the AMP algorithm. It is straightforward to derive this characterization from (70,71):

$$\hat{s}_{i\to\mu}^{t+1} = G_s(\lambda_h^t, \sum_{\nu\neq\mu} \lambda_h^t \alpha_{\nu\to i}^t X_{\nu i}), \tag{88}$$

$$\hat{\alpha}_{\mu\to i}^t = -G_y(\lambda_\eta^t, y_\mu, \sum_{j\neq i} X_{\mu j} \hat{s}_{j\to\mu}^t). \tag{89}$$

It is possible to track this algorithm using only a few scalar values as we now explain. From (88) it follows that $\hat{s}_{j\to\mu}^t$ is independent of $X_{\mu j}$, thus from the central limit theorem (CLT) the sum $\sum_{j\neq i} X_{\mu j} \hat{s}_{j\to\mu}^t$ approaches a Gaussian of the same variance for any index $\mu$ in the large system limit. We therefore define $\eta^t$ to be a Gaussian random variable with the same variance so that the update equation for $\lambda_h$ (73) becomes:

$$\lambda_h^t = \left( \alpha\gamma \left\langle \frac{\partial}{\partial\eta} G_y(\lambda_\eta^t, y, \eta^t) \right\rangle \right)^{-1}. \tag{90}$$

Similarly, for each index $i$, $\sum_{\nu\neq\mu} \lambda_h^t \alpha_{\nu\to i}^t X_{\nu i}$ will (by CLT) approach a Gaussian random variable $h^t$ with the same mean and variance for any $i$ in the large system limit:

$$h^t = -\sum_{\nu\neq\mu} X_{\nu i} \lambda_h^t G_y(\lambda_\eta^t, y_\nu, \sum_{j\neq i} X_{\nu j} \hat{s}_{j\to\nu}^t). \tag{91}$$

We now compute the mean and standard deviation of $h^t$ using the fact that the outputs are drawn from a model $y = r(z, \epsilon)$. We track this mean and standard deviation by defining scalars $\mu^t$ and $q_h^t$:

$$\mu^t s^0 = \langle h^t \rangle = -\alpha\gamma s^0 \left\langle \lambda_h^t \frac{\partial}{\partial z} G_y(\lambda_\eta^t, r(z, \epsilon), \eta^t) \right\rangle_{z, \eta^t, \epsilon}, \tag{92}$$

$$q_h^t = \langle (\delta h^t)^2 \rangle = \alpha\gamma \left\langle \left( \lambda_h^t G_y(\lambda_\eta^t, r(z, \epsilon), \eta^t) \right)^2 \right\rangle_{z, \eta^t, \epsilon}. \tag{93}$$

Since the measurement matrix and true parameter values are drawn independently, $\mathbf{x}_\mu \cdot \mathbf{s}^0$ approaches a Gaussian random variable of variance $\gamma\sigma_s^2$ for any $\mu$; $z$ is defined to be the same Gaussian random variable. Similarly, the other message passing equations yield a single letter characterization:

$$\lambda_\eta^{t+1} = \gamma\lambda_h^t \left\langle \frac{\partial}{\partial h} G_s(\lambda_h^t, h^t) \right\rangle_h^t, \tag{94}$$

$$\hat{s}^{t+1} = G_s(\lambda_h^t, h^t), \tag{95}$$

$$C_s^{t+1} = \text{cov}(s^{t+1}, s^0). \tag{96}$$

Here $C_s$ tracks the covariance between the parameter estimates and true parameters so that from their definition, $\eta^t$ and $z$ are also zero mean, correlated Gaussian variables with covariance:

$$C_\eta^t = \text{cov}(\eta^t, z) = \gamma\text{cov}(s^t, s^0). \tag{97}$$

## 3.2 bAMP state evolution

The general SE equations derived in the previous section simplify in the case of bAMP. One of the reasons for this is that $\mu^t = 1$ for each update time step $t$, as is shown in [7]. From (92), proving $\mu^t = 1$ can be accomplished to proving that

$$- \alpha\gamma\lambda_h^t \big\langle \tfrac{\partial}{\partial z} G_y^B(\lambda_\eta^t, r(z,\epsilon), \eta^t) \big\rangle_{z,\eta^t,\epsilon} = 1, \tag{98}$$

and

$$\big\langle (\hat{s}^t)^2 \big\rangle = \big\langle \hat{s}^t s^0 \big\rangle = \sigma_s^2 - \frac{1}{\gamma}\lambda_\eta^t. \tag{99}$$

*Derivation —.*

The claim follows by induction, assuming that at we initialize $\lambda_\eta = \gamma\sigma_s^2$ and $\hat{s}_j = 0$, so that (99) is true for the base case $t = 0$. For the inductive hypothesis we assume (99) holds up to iteration $t$, and show that (98) also holds at iteration $t$, and that (99) holds for $t + 1$. We begin by using the definition of $G_y^B$ to show:

$$\big\langle \eta G_y^B(\lambda_\eta, r(z,\epsilon), \eta) \big\rangle_{z,\eta,\epsilon} \tag{100}$$

$$= - \Big\langle \eta \tfrac{\partial}{\partial \eta} \log \Big( \int P_y(y|z') e^{-\frac{(\eta-z')^2}{2\lambda_\eta}} dz' \Big) \Big\rangle_{\eta,z,y} \tag{101}$$

$$= - \Big\langle \eta \tfrac{\partial}{\partial \eta} \int P_y(y|z) e^{-\frac{(\eta-z)^2}{2\lambda_\eta}} dz\, dy \Big\rangle_\eta = 0. \tag{102}$$

The final equality follows from the fact that the integral contained in the average is a constant so that its derivative is zero. We next apply Stein's lemma, which can be derived from integration by parts and implies that any pair of zero mean correlated Gaussian random variables $x, y$ satisfy the relation:

$$\big\langle g(x)y \big\rangle = \big\langle g'(x) \big\rangle \mathrm{cov}(x, y). \tag{103}$$

$z$ and $\eta$ are two such correlated Gaussian random variables and it follows that from the inductive hypothesis (99) as well as (97) that $\mathrm{cov}(\eta, z) = \big\langle \eta^2 \big\rangle$. Applying Stein's lemma to (100) implies:

$$0 = \big\langle \eta G_y^B(\lambda_\eta, r(z,\epsilon), \eta) \big\rangle \tag{104}$$

$$\propto \big\langle \tfrac{\partial}{\partial z} G_y^B(\lambda_\eta, r(z,\epsilon), \eta) \big\rangle + \big\langle \tfrac{\partial}{\partial \eta} G_y^B(\lambda_\eta, r(z,\epsilon), \eta) \big\rangle. \tag{105}$$

Multiplying both sides of the result above by $\alpha\gamma\lambda_h^t$, it follows that

$$- \alpha\gamma\lambda_h^t \big\langle \tfrac{\partial}{\partial z} G_y^B(\lambda_\eta, r(z,\epsilon), \eta) \big\rangle = \alpha\gamma\lambda_h^t \big\langle \tfrac{\partial}{\partial \eta} G_y^B(\lambda_\eta, r(z,\epsilon), \eta) \big\rangle = 1, \tag{106}$$

where the final equality is a result of SE relation (90). This proves (98), implying that $\mu^t = 1$. It is also helpful to now show that for bAMP SE, $q_h^t = \lambda_h^t$:

$$\lambda_h^t = \Big( \alpha\gamma \big\langle \big( \tfrac{\partial}{\partial \eta} G_y^B(\lambda_\eta^t, y, \eta^t) \big)^2 \big\rangle \Big)^{-1} \tag{107}$$

$$= \big( \alpha\gamma J\,[\,P_y(y, \eta^t, \lambda_\eta^t)\,] \big)^{-1}. \tag{108}$$

From (93) $q_h^t$ has the form

$$q_h^t = \alpha\gamma(\lambda_h^t)^2 \big\langle \big( \partial_\eta \log(P_y(y, \eta^t, \lambda_\eta^t)) \big)^2 \big\rangle \tag{109}$$

$$= \big( \alpha\gamma J\,[\,P_y(y, \eta^t, \lambda_\eta^t)\,] \big)^{-1} = \lambda_h^t. \tag{110}$$

It then follows from (95) that the updated $\hat{s}$ the posterior mean given the underlying parameter value corrupted by a Gaussian random variable of variance: $s_{\lambda_h}^0 = s^0 + \sqrt{\lambda_h} z$. Thus,

$$\hat{s}^{t+1} = \hat{s}^{\mathrm{MMSE}}(s_{\lambda_h^t}). \tag{111}$$

The fact that the algorithm is performing MMSE inference under corruption allows us to show that

$$\left\langle \, \hat{s}(s^0_{\lambda_h}) s^0 \, \right\rangle_{s^0, s^0_{\lambda_h}} = \left\langle \, \hat{s}^2(s^0_{\lambda_h}) \, \right\rangle_{s^0_{\lambda_h}}. \tag{112}$$

This follow from the fact that

$$\hat{s}(s^0_{\lambda_h}) = \int s^0 P(s^0 | s^0_{\lambda_h}) ds^0, \tag{113}$$

therefore the LHS of (112) becomes:

$$\left\langle \, \hat{s}(s^0_{\lambda_h}) s^0 \, \right\rangle_{s^0, s^0_{\lambda_h}} = \int P(s^0_{\lambda_h}) ds^0_{\lambda_h} \int s^0 P(s^0 | s^0_{\lambda_h}) ds^0 \int s P(s | s^0_{\lambda_h}) ds \tag{114}$$

$$= \int P(s^0_{\lambda_h}) ds^0_{\lambda_h} \left( \int s^0 P(s^0 | s^0_{\lambda_h}) ds^0 \right)^2 = \left\langle \, \hat{s}^2(s^0_{\lambda_h}) \, \right\rangle_{s^0_{\lambda_h}}. \tag{115}$$

It follows that the (99) holds at iteration $t+1$, which completes the derivation.—

We now use the fact that $\mu^t = 1$ for all iterations to demonstrate that the SE equations for bAMP simplify to:

$$\lambda^t_h = (\alpha\gamma \langle \, \partial_\eta G^B_y(\lambda^t_\eta, y, \eta^t) \, \rangle)^{-1}, \qquad \lambda^{t+1}_\eta = \gamma\lambda^t_h \left\langle \, \partial_s G^B_s(\lambda^t_h, s^0 + \sqrt{q^t_h} w) \, \right\rangle, \tag{116}$$

$$q^t_h = \alpha\gamma \left\langle \, \left( \lambda^t_h G^B_y(\lambda^t_\eta, y, \eta^t) \right)^2 \, \right\rangle, \qquad q^{t+1}_\eta = \gamma \left\langle \, \left( G^B_s(\lambda^t_h, s^0 + \sqrt{q^t_h} w) - s^0 \right)^2 \, \right\rangle. \tag{117}$$

Here $q^t_\eta$ is the variance of the components of the residual $\eta^t - z$. It then follows from induction and the form of $G^B_y$ and $G^B_s$ that $q^t_\eta = \lambda^t_\eta$ and $q^t_h = \lambda^t_h$, so that the following pair of update equations fully describe the state evolution of the system:

$$\lambda^t_h = \frac{1}{\alpha\gamma J \left[ P_y(y, \eta^t, \lambda^t_\eta) \right]}, \qquad \lambda^{t+1}_\eta = \gamma \mathrm{MMSE}(s^0 | s^0 + \sqrt{\lambda^t_h} w). \tag{118}$$

The fact that $q^t_s = \left\langle \, (\hat{s}^t - s^0)^2 \, \right\rangle = \frac{q^t_\eta}{\gamma}$ follows from (97).

# 4 Connection between bAMP and mAMP

For bAMP the message passing equations are simply:

$$\hat{s}^{t+1}_{i \to \mu} = G^B_s(\lambda^t_h, \sum_{\nu \neq \mu} \lambda^t_h \alpha^t_{\nu \to i} X_{\nu i}), \tag{119}$$

$$\alpha^t_{\mu \to i} = -G^B_y(\lambda^t_\eta, y_\mu, \sum_{j \neq i} X_{\mu j} \hat{s}^t_{j \to \mu}). \tag{120}$$

For mAMP the message passing equations are:

$$s^{t+1}_{i \to \mu} = \mathcal{P}_{\lambda^t_h}[\sigma](\lambda^t_h \sum_{\nu \neq \mu} \alpha^t_{\nu \to i} X_{\nu i}), \tag{121}$$

$$\alpha^t_{\mu \to i} = -\mathcal{M}_{\lambda^t_\eta}[\mathcal{L}(y_\mu, \cdot)]'(\sum_{j \neq i} X_{\mu j} s^t_{j \to \mu}). \tag{122}$$

Thus the two results are equivalent under the choice

$$\mathcal{M}_{\lambda_\eta}[\mathcal{L}(y, \cdot)](\eta) = -\log \left( \int P_y(y|z) e^{-\frac{(\eta - z)^2}{2\lambda_\eta}} dz \right), \tag{123}$$

$$\mathcal{P}_{\lambda_h}[\sigma](h) = h + \lambda_h \frac{\partial}{\partial h} \log \left( \int P_s(s) e^{-\frac{(h - s)^2}{2\lambda_h}} ds \right). \tag{124}$$

Equation (124) can be written in terms of the Moreau envelope as:

$$\mathcal{M}_{\lambda_h}[\sigma](z) = -\log \left( \int P_s(s) e^{-\frac{(z - s)^2}{2\lambda_h}} ds \right). \tag{125}$$

To see why, apply the relation between the proximal map and Moreau envelope from appendix A.1 to (124) and integrate the form which contains a derivative of the Moreau envelope, noting that the additive constant introduced by integration may be neglected because it will not alter the performance of an M-estimator.

In order to compute the information theoretically optimal M-estimator $\mathcal{L}^{\mathrm{opt}}, \sigma^{\mathrm{opt}}$ with the same fixed points as bAMP, we first compute the fixed points of bAMP SE $\lambda_\eta, \lambda_h$ using (118). Under this choice of $\lambda_\eta, \lambda_h$ it is possible to invert $\mathcal{M}_\lambda[\,f\,](x) = g$, under the assumption that $g$ is convex (see appendix A.3), which will certainly hold under if $P_y(y|z)$ is log-concave in $z$ and $P_s$ is log-concave, since the Gaussian distribution is also log-concave, and because log-concavity is preserved under convolutions. Applying the formula derived in appendix A.3 for inverting the Moreau envelope, the forms of the optimal loss and regularization functions are:

$$\mathcal{L}^{\mathrm{opt}}(y, \eta) = -\mathcal{M}_{\lambda_\eta}\left[\log\left(\int P_y(y|z)e^{-\frac{(\cdot-z)^2}{2\lambda_\eta}}\,dz\right)\right](\eta), \tag{126}$$

$$\sigma^{\mathrm{opt}}(h) = -\mathcal{M}_{\lambda_h}\left[\log\left(\int P_s(s)e^{-\frac{(\cdot-s)^2}{2\lambda_h}}\,ds\right)\right](h). \tag{127}$$

## 5 Examples and special cases

### 5.1 Additive noise:

Optimal M-estimation has been derived using different (variational) methods in the case of additive noise [8, 9]. In this scenario, output data is drawn according to the model:

$$y_\mu = \mathbf{x}_\mu \cdot \mathbf{s}^0 + \epsilon_\mu \tag{128}$$

In the non-additive noise case we have that optimal loss function of two variables $\mathcal{L}(y, \eta)$. It takes the form:

$$\mathcal{M}_{\lambda_\eta}[\mathcal{L}(y, \cdot)](\eta) = -\log\left(\int P_y(y|z)e^{-\frac{(\eta-z)^2}{2\lambda}}\,dz\right), \tag{129}$$

however in the case of linear noise, previous work has considered only a single variable loss function. In the linear noise case we can write our loss function as a single variable by replacing $\mathcal{L}(y, \cdot)$ by $\rho(y - \cdot)$. Note that under additive noise one can also replace $P_y(y|z)$ by $P_\epsilon(y - z)$. Under this change of variable the previous relation may be written as

$$\mathcal{M}_{\lambda_\eta}[\rho](y - \eta) = -\log\left(\int P_\epsilon(y - z)e^{-\frac{(\eta-z)^2}{2\lambda_\eta}}\,dz\right) = -\log\left(\int P_\epsilon(\epsilon)e^{-\frac{(\epsilon-y+\eta)^2}{2\lambda_\eta}}\,d\epsilon\right). \tag{130}$$

It then follows, see appendix A.4 for a derivation, that the optimal penalty $\rho$ satisfies:

$$\mathcal{M}_{\lambda_\eta}[\rho](z) = -\log\left(\int P_\epsilon(\epsilon)e^{-\frac{(z-\epsilon)^2}{2\lambda_\eta}}\,d\epsilon\right), \tag{131}$$

which recovers the findings of the variational approaches [8, 9].

### 5.2 Logistic regression

Here we consider logistic regression as an example of non-additive noise. We derive the loss function corresponding to maximum likelihood, which is optimal in the classical setting. We then define $z_\mu = \mathbf{x}_\mu \cdot \mathbf{s}^0$, and let the probability that $y_\mu = 1$ be

$$h(z_\mu) = \frac{1}{1 + e^{-z_\mu}}, \tag{132}$$

and let $y_\mu = 0$ otherwise. It follows that, the probability of a measuring a vector $\mathbf{y}$ may be written as

$$P(\mathbf{y}|\boldsymbol{z}) = \prod_\mu h(z_\mu)^{y_\mu}(1 - h(z_\mu))^{1-y_\mu} = \exp\left[\sum_\mu y_\mu \log(h(z_\mu)) + (1 - y_\mu)\log(1 - h(z_\mu))\right], \tag{133}$$

which may be rearranged in the simpler form:

$$P(\mathbf{y}|\boldsymbol{z}) = \exp\left[\sum_\mu y_\mu z_\mu + \sum_\mu \log\left(\frac{1}{e^{z_\mu}+1}\right)\right]. \tag{134}$$

ML corresponds to maximizing the probability above, or equivalently minimizing the negative logarithm of the previous expression:

$$\hat{\mathbf{s}}^{\mathrm{ML}} = \arg\min_{\mathbf{s}} \sum_\mu \left(-y_\mu \mathbf{x}_\mu \cdot \mathbf{s} + \log\left(e^{\mathbf{x}_\mu \cdot \mathbf{s}}+1\right)\right). \tag{135}$$

Therefore the loss function $L(y_\mu, \eta_\mu)$ corresponding to ML has the form

$$L(y_\mu, \eta_\mu) = -y_\mu \eta_\mu + \log\left(e^{\eta_\mu}+1\right), \tag{136}$$

and for ML, there is no regularization ($\sigma = 0$). It is not difficult to show that this optimization problem is convex in $\boldsymbol{\eta}$, so that the output channel is log-concave and our optimal loss function derivation is justified for this model.

# 6  Large sparse system limit

BP is exact on trees, and when the measurement matrices are sufficiently sparse, for instance when the number of non-zero elements grows as $\log(N)$, then the corresponding factor graph (on which BP is performed) becomes locally tree like so that loops shorter than any finite length have a vanishingly low probability for sufficiently large $N$. The lack of loops implies that the BP equations (24,25) will be exact for sufficiently sparse measurement matrices. One such assumption we could make, as in [10] is to let the fraction of non-zero elements in a measurement $\mathbf{x}_\mu$ be $f$ where $\lim_{P\to\infty} fP = \infty$ and $\lim_{P\to\infty} fP^a = 0$ for $a < 1$. We let the non-zero values of the measurement matrix be drawn iid from a distribution $P_x$ with zero mean, variance $\frac{\gamma}{f}$, and finite $4^{\mathrm{th}}$ order moment. The requirement that $\lim_{P\to\infty} fP = \infty$ is needed for central limit theorem to apply throughout the derivation including SE and for the assumption in 2.2 and 2.3 that only the first 2 terms in the Taylor expansion in the messages need to be kept to recover BP (i.e. $\sum_i X_{\mu i}^3 \to 0$). Under these assumptions, BP is provably exact and equivalent to AMP so that both bAMP and mAMP are correct. For a rigorous treatment of the large sparse limit see [11] which proves the correctness of bAMP in the large sparse limit or [12] which does the same for SE.

# A  Useful properties of the Moreau envelope and proximal map

The Moreau envelope is a functional map and maps a function $f$ to

$$\mathcal{M}_\lambda[\,f\,](x) = \min_y \left[\frac{(x-y)^2}{2\lambda} + f(y)\right], \tag{137}$$

where $\lambda$ is a scalar parameterizing the mapping and we will denote the special case of $\mathcal{M}_1[f]$ by $\mathcal{M}[\,f\,]$. Some properties that follows from this definition are that the minimizers of $f$ and $\mathcal{M}_\lambda[\,f\,]$ are the same, and that the Moreau envelope is a lower bound on the function $f$. A related function, called the proximal map is defined as

$$\mathcal{P}_\lambda[\,f\,](x) = \arg\min_y \left[\frac{(x-y)^2}{2\lambda} + f(y)\right]. \tag{138}$$

## A.1  Relation between proximal map and Moreau envelope

The proximal map can be viewed as performing a gradient descent step along the Moreau envelope:

$$\mathcal{P}_\lambda[\,f\,](x) - x = -\lambda \mathcal{M}_\lambda[\,f\,]'(x). \tag{139}$$

To derive this result we differentiate the Moreau envelope, performing the minimization before the differentiation:

$$\mathcal{M}_\lambda[f]'(x) = \frac{d}{dx} \min_y \left[ \frac{(x-y)^2}{2\lambda} + f(y) \right] = \frac{d}{dx} \left[ \frac{(x-\hat{y})^2}{2\lambda} + f(\hat{y}) \right], \qquad (140)$$

where $\hat{y}$ is the minimizer of the RHS argument of (140). Differentiating with respect to $\hat{y}$ yields $0$ at the minimum, so the $\hat{y}$ may be effectively treated as a constant and we need only differentiate with respect to $x$, which yields

$$\mathcal{M}_\lambda[f]'(x) = \frac{x-\hat{y}}{\lambda}. \qquad (141)$$

It follows that

$$\mathcal{M}_\lambda[f]'(x) = \frac{x - \mathcal{P}_\lambda[f](x)}{\lambda}. \qquad (142)$$

## A.2 Relation between proximal map and derivative

$$x - \mathcal{P}_\lambda[f](x) = \lambda f'(\mathcal{P}_\lambda[f](x)). \qquad (143)$$

The result follows from the fact that $\mathcal{P}_\lambda[f](x)$ is defined to be a minimizer of

$$F(x, y) = \frac{(x-y)^2}{2\lambda} + f(y) \qquad (144)$$

with respect to $y$, and thus for differentiable $f$, $\frac{\partial}{\partial y} F(x, y) = 0$ under the choice $y = \mathcal{P}_\lambda[f](x)$. Since

$$\frac{\partial}{\partial y} F(x, y) = \frac{y-x}{\lambda} + f'(y), \qquad (145)$$

substituting $y = \mathcal{P}_\lambda[f](x)$ and requiring the result to be equal to zero yields the desired result.

## A.3 Inverse of the Moreau envelope

*For $\lambda > 0$ and $f$ a convex, lower semi-continuous function such that $\mathcal{M}_\lambda[f] = g$, the Moreau envelope can be inverted so that $f = -\mathcal{M}_\lambda[-g]$.*

*Derivation–.*

To derive this result, we first consider the case of $\lambda = 1$, from which the $\lambda > 0$ case will follow. Our assumption that $\mathcal{M}_\lambda[f] = g$ implies

$$g(x) = \mathcal{M}[f](x) = \min_y \left[ \frac{(x-y)^2}{2} + f(y) \right] = \frac{x^2}{2} + \min_y \left[ -xy + \frac{y^2}{2} + f(y) \right] = \frac{x^2}{2} - \max_y \left[ xy - \frac{y^2}{2} - f(y) \right].$$
$$(146)$$

We now define the Fenchel conjugate $.^*$, which operates on a function $h$ to yield $h^*(x) = \max_y [xy - h(y)]$. We then define, for notational simplicity, the function $p_2(x) = \frac{x^2}{2}$. With this notation, (146) reduces to

$$g = p_2 - (f + p_2)^*. \qquad (147)$$

The Fenchel-Moreau theorem [13] states that if $h$ is a convex and lower semi-continuous function, then $h = (h^*)^*$. These properties are assumed true for $f$ and will also hold for $f + p_2$ so that (147) may be inverted, yielding:

$$f = (p_2 - g)^* - p_2. \qquad (148)$$

We now write $f$ in terms of a Moreau envelope by expanding the previous expression:

$$f(x) = -\frac{x^2}{2} + \max_y \left[ xy - \frac{y^2}{2} + g(y) \right] = -\min_y \left[ \frac{(x-y)^2}{2} - g(y) \right] = -\mathcal{M}[-g](x). \qquad (149)$$

Thus, $\mathcal{M}[f] = g$ implies $f = -\mathcal{M}[-g]$. To extend this to $\lambda \neq 1$, we use the identity

$$\lambda \mathcal{M}_\lambda[\tfrac{1}{\lambda} f] = \mathcal{M}[f], \qquad (150)$$

which can be verified by substitution into the definition of the Moreau envelope (137). Combining the result $\mathcal{M}[f] = g$ implies $f = -\mathcal{M}[-g]$ with (150) yields that:

$$\mathcal{M}_\lambda[\tfrac{1}{\lambda}f] = \frac{1}{\lambda}g, \tag{151}$$

also implies

$$\frac{1}{\lambda}f = -\mathcal{M}_\lambda[-\tfrac{1}{\lambda}g], \tag{152}$$

which completes the derivation since $\frac{1}{\lambda}$ may be absorbed into the definition of $f$ and $g$.

### A.4 Moreau envelope for additive noise model

Under an additive noise model, we consider loss functions of the form $\mathcal{L}(y, z) = \rho(y - z)$. Under this setting we show that

$$\mathcal{M}_{\lambda_\eta}[\mathcal{L}(y, \cdot)](x) = \mathcal{M}_{\lambda_\eta}[\rho](y - x). \tag{153}$$

This relations follows from the definition of the Moreau envelope:

$$\mathcal{M}_{\lambda_\eta}[\mathcal{L}(y, \cdot)](x) = \mathcal{M}_{\lambda_\eta}[\rho(y - \cdot)](x) = \min_z \left[ \frac{(x - z)^2}{2\lambda_\eta} + \rho(y - z) \right]. \tag{154}$$

Thus, under the change of variables $w = y - z$, the above expression equals

$$\min_z \left[ \frac{(y - x - w)^2}{2\lambda_\eta} + \rho(w) \right] = \mathcal{M}_{\lambda_\eta}[\rho](y - x). \tag{155}$$