[Reviews · NeurIPS 2016]

Reviewer 1

Summary

The authors argue for an equivalence between regularized MAP/M-estimation problems and Bayes optimal inference in high-dimensional inference. While MAP/M-estimation is a widely used strategy in high-dimensional inference, there is in general no guarantee that the resulting estimators/methods are Bayes optimal -- Bayes optimality (under squared-error loss) is obtained via the posterior mean, not mode/maximization. The authors argue that under an appropriate high-dimensional limit, there is actually a correspondence between MAP/M-estimation problems and Bayes optimal inference. To establish this correspondence, the authors rely on approximate message passing (AMP) techniques.

Qualitative Assessment

I think the results are extremely interesting, and could have wide impact. However, I'm somewhat uncomfortable with the lack of clearly stated rigorous results in the paper. This lack of clarity leads me to question which of the claimed results/equivalences are in fact rigorously derived and which are heuristics. Elsewhere in the AMP literature, rigorous theoretical results often rely on pretty strong distributional assumptions. However, there are also a variety of heuristics and conjectures in AMP for more general settings/relaxed distribution assumptions. Where on this spectrum of rigor does this paper fall?

Confidence in this Review

2-Confident (read it all; understood it all reasonably well)


Reviewer 2

Summary

The paper demonstrates how MMSE (minimum mean squared error) performance is asymptotically achievable via optimization technique, with an appropriately selected convex penalty and regularization function, which are a smoothed version of MAP (maximum a posteriori) algorithm. The paper provides a derivation for optimal M-estimators proposed by El Karoui, et. al. PNAS 2013. The main focus of this paper is to answer: (1) How does the performance of the optimal M-estimators as derived in El Karoui, et. al. PNAS 2013 and Bayati & Montanari 2011 compare to that of Bayesian inference? (2) What is the form of the optimal M-estimator under a non-additive noise model? The authors answer these questions with a form of generalized approximate message passing (AMP) algorithm, which is GAMP algorithm. The authors provide simulations using a non-additive noise model which demonstrate a substantial improvement in accuracy achieved by optimal M-estimation over MAP and discuss the model conditions under which the optimal M-estimator will be Bayes optimal.

Qualitative Assessment

The paper is well written and well presented! The authors are able to establish the novelty of their approach with significant theoretical justification. Certainly, the paper is theoretically strong with proofs and derivations clearly laid out. It would be good if the authors can elaborate on the computational complexity of using their M-estimation technique compared to MAP estimators. Only concern of their approach is the practical applicability, as the paper demonstrate "performance of M-estimation for finite sized datasets with N and P each on the order of a couple hundred". It would be nice if the authors could provide some real life application of their approach and hence the superiority over MAP.

Confidence in this Review

2-Confident (read it all; understood it all reasonably well)


Reviewer 3

Summary

The authors demonstrate an equivalence of high dimensional Bayes optimal inference and M-estimators through an appropriate choice of regularization functions.

Qualitative Assessment

1. This paper is very difficult to follow. The sections are essentially large blobs of texts with high amount of notations and abbreviations. The discussions in sections 2 and 3 are terse and without proper references. Section 4 is not well explained either. 2. It is not clear as to how a practitioner should utilize the equivalence of the two paradigms. 3. It would have helped to see some applications in synthetic or real data.

Confidence in this Review

1-Less confident (might not have understood significant parts)


Reviewer 4

Summary

The goal of this paper is to find a solution to the minimum mean squared error problem (MMSE) by establishing an equivalence between high-dimensional Bayes integral and high-dimensional convex optimization. This paper addresses the open issues of comparing optimal Bayesian inferences with optimal M-estimators and investigates the class of M-estimators for non-additive noise models. The authors consider the modern high-dimensional set-up where N,P both diverge keeping the ratio N/P fixed, an area that has received a lot of attention in the recent past. The connection between Bayes optimal inference and M-estimators is derived analytically through approximate message passing algorithm.

Qualitative Assessment

I am not an expert in this area but I liked the author’s approach of connecting M-estimation and Bayes optimal inference through the fixed points of bAMP and mAMP, and I believe this should lead to some advancement in this area through the extension to non-additive noise. I particularly like the correspondence between MAP inference and M-estimators through the smoothing terms. This approach has also the added advantage of adapting to the high-dimensional (and potentially sparse parameters through approximating log-concave parameter distribution) situation as well as the low dimensional situation through the measurement density and certain parameters going to zero. A minor question in this context: the authors stated that the M-estimation will be optimal in the case of a flat parameter distribution such as a Gaussian with very high variance – I am curious to know how it behaves if the parameter distribution is heavy-tailed (such as Cauchy) instead, as some authors in Bayesian literature suggest using Cauchy as an alternative to normal with high variance as non-informative priors (Gelman, A (2006) & Bhadra et al. (2016)). Although I admit this might be beyond the scope of the current discussion. References: Gelman, A. (2006) Prior distributions for variance parameters in hierarchical models, Bayesian Analysis. Bhadra et al. (2016) Default Bayesian Analysis with Global-Local Shrinkage Priors, arXiv preprint arXiv:1510.03516

Confidence in this Review

1-Less confident (might not have understood significant parts)


Reviewer 5

Summary

The paper shows the equivalence between Bayes optimal inference and solving certain M-estimators for inference in high-dimensional regression. In particular they show that solutions of smoothed versions of convex M-estimators like Lasso are equivalent to the Bayes optimal estimator.

Qualitative Assessment

The paper will be of interest to people working on the theoretical aspects of Bayesian inference for high dimensional regression. But I have reservations regarding its practical applicability. First the results hold in the asymptotic case with not much clarity on the performance in the non-asymptotic case. Second the experimental results are on simulated data assuming the parameters are from a Laplacian distribution. For truly sparse vectors or real world problems where it is sometimes beneficial to infer the most important variables the assumption of sampling from a Laplacian distribution may not be true. It will have been more convincing if the performance of the M-estimators like Lasso and their smoothed versions presented in this paper were compared on real world datasets. That said the authors do mention this as part of future and in that sense I believe it should be a good introduction to such estimators for the conference participants.

Confidence in this Review

1-Less confident (might not have understood significant parts)